# Estimating aerosol emission from SPEXone on the NASA PACE mission using an ensemble Kalman Smoother: Observing System Simulation Experiments (OSSEs)

Athanasios Tsikerdekis [1,2], Nick A.J Schutgens [2], Guangliang Fu [1], Otto P. Hasekamp [1]

[1]SRON Netherlands Institute for Space Research, Leiden, the Netherlands
[2]Department of Earth Science, Vrije Universiteit Amsterdam, 1081 HV Amsterdam, the Netherlands

*Correspondence to*: Athanasios Tsikerdekis (A.Tsikerdekis@sron.nl)

**Abstract.** We present a top-down approach for aerosol emission estimation from SPEXone polarimetric retrievals related to the aerosol amount, size, and absorption using a fixed-lag ensemble Kalman smoother (LETKS) in combination with the
ECHAM-HAM model. We assess the system by performing Observing System Simulation Experiments (OSSEs), in order to evaluate the ability of the future multi-angle polarimeter instrument, SPEXone, as well as a satellite with near perfect global coverage. In our OSSEs, the Nature Run (NAT) is a simulation by the global climate aerosol model ECHAM-HAM with altered aerosol emissions. The Control (CTL) and the data assimilation (DAS) experiments are composed of an ensemble of ECHAM-HAM simulations, where the default aerosol emissions are perturbed with factors taken from a Gaussian distribution.
Synthetic observations, specifically Aerosol Optical Depth at 550nm ($AOD_{550}$), Angstrom Exponent from 550nm to 865nm ($AE_{550-865}$) and Single Scattering Albedo at 550nm ($SSA_{550}$) are assimilated in order to estimate the aerosol emission fluxes of desert dust (DU), sea salt (SS), organic carbon (OC), black carbon (BC) and sulphate ($SO_4$), along with the emission fluxes of two $SO_4$ precursor gases ($SO_2$, DMS). The prior emission global relative Mean Absolute Error (MAE) before the assimilation ranges from 33% to 117%. Depending on the species, the assimilated observations sampled using the satellite with near perfect
global coverage, reduce this error to equal to or lower than 5%. Despite its limited coverage, the SPEXone sampling bares similar results, with somewhat larger errors for DU and SS (both having a MAE equal to 11%). Further, experiments show that doubling the measurement error, increases the global relative MAE up to 22% for DU and SS. In addition, our results reveal that when the wind of DAS uses a different reanalysis dataset (ERA5 instead of ERA-interim) than the NAT, the estimated SS emissions are negatively affected the most, while other aerosol species are negatively affected to a smaller extent.
If the DAS uses dust or sea salt emission parametrisations that are very different from the NAT, posterior emissions can still be successfully estimated, but this experiment revealed that the source location is important for the estimation of dust emissions. This work suggests that the upcoming SPEXone sensor will provide observations related to aerosol amount, size and absorption with sufficient coverage and accuracy, in order to estimate aerosol emissions.

# 1 Introduction

Data assimilation methods can greatly improve the aerosol representation in the atmosphere by combining the simulated aerosol state of a model with the observed aerosol optical and microphysical properties retrieved from satellites. The accuracy of the spatiotemporal distribution of an aerosol species in a data assimilation product depends both on the accuracy of the simulated processes in the model as well as the quality and the type of the assimilated observations. Several past studies estimated aerosol emission based on remote sensing observations (Dubovik et al., 2008; Jin et al., 2019; Pope et al., 2016;

Sekiyama et al., 2010; Xu et al., 2013), although only some studies assimilated size related measurements, such as Aerosol Optical Depth (AOD) in two wavelengths or fine and coarse AOD or Angstrom Exponent (AE) (Escribano et al., 2017; Huneeus et al., 2012; Schutgens et al., 2012). In addition, very few recent studies assimilated absorption related measurements like Absorption Aerosol Optical Depth (AAOD) or Single Scattering Albedo (SSA) to correct either the aerosol mixing ratio (Tsikerdekis et al., 2021) or the aerosol emissions (Chen et al., 2018, 2019). Absorption observations were used by

Kacenelenbogen et al. (2019) to estimate the short-wave direct aerosol effect from the A-Train satellite sensors. Further, Schutgens et al. (2021) intercompared and evaluated with AERONET four satellite products (FL-MOC, OMAERUV, POLDER-GRASP and POLDER-SRON) for AAOD and SSA and suggested that satellite absorption observations could be used to evaluate AEROCOM model biases, since the diversity of model biases is larger than satellite biases.

It has been noted in the past that multi-viewing angle and multi-wavelength intensity and polarization measurements with high

accuracy have the largest capability to provide the aerosol properties relevant to climate research (Hasekamp and Landgraf, 2007). Recently, Hasekamp et al. (2019b) showed that polarimetric satellite retrievals related to aerosol shape, size and number provide a more accurate aerosol indirect radiative effect compared to previous observational-based studies. Unfortunately only one such Multi-Angle Polarimeter (MAP) provided aerosol optical and microphysical properties from space for several years in the past (2004-2013), the Polarization and Directionality of Earth Reflectances (POLDER-3) on board of the microsatellite

PARASOL (Dubovik et al., 2019).

Several MAP instruments are scheduled for launch in the coming 3 years (Dubovik et al., 2019), with the NASA PACE mission (Werdell et al., 2019) hosting two MAP sensors onboard, the Spectropolarimeter for Planetary Exploration SPEXone (Hasekamp et al., 2019a) and the Hyper-Angular Rainbow Polarimeter-2 (HARP-2). Since these instruments are not yet in space, their observational capabilities on aerosol optical properties and consequently, their potential to estimate aerosol

species-specific emission fluxes, can only be theoretically predicted with Observing System Simulation Experiments (OSSEs) (Arnold and Dey, 1986; Timmermans et al., 2015). In OSSEs a model simulation is assumed as reality, also known as the Nature Run (NAT), from which synthetic measurements are sampled based on the spatiotemporal coverage of an assumed satellite sensor. Subsequently two experiments are conducted, a control (CTL) and a data assimilation (DAS) experiment, in which the sampled synthetic observations from the NAT are assimilated. Note that the NAT and the CTL simulations are

different experiments, either by using a totally different model or by using the same model with different emissions or/and

physics options. The ability of the instrument to estimate the aerosol state can be highlighted by evaluating the CTL and the DAS experiments with NAT.

Timmermans et al. (2008) firstly used OSSEs with an ensemble Kalman filter to assess the ability of assimilated AOD sampled based on an imager type instrument and assimilated PM2.5 sampled based on the location of ground based stations, with the goal to estimate PM2.5 concentrations over Europe. Meland et al. (2013) used OSSEs with an adjoint inverse data assimilation method for aerosol emission estimation to assess the benefits of remote polarimetric over intensity measurements. Even though the intensity measurements had broader spatial coverage, aerosol emissions were 3 times more sensitive to the polarized reflectance at the top of the atmosphere compared to the radiant reflectance at the top of the atmosphere. Further it was highlighted that assimilated multi-angle polarimetric measurements could substantially improve aerosol simulations. Subsequent studies using real POLDER retrievals confirmed this for aerosol mixing ratio estimation (Tsikerdekis et al., 2021) and aerosol emission estimation (Chen et al., 2019) from the POLDER-3 instrument. Yumimoto and Takemura (2013) used OSSEs and a 4D-Var data assimilation system to estimate aerosol emissions based on simulated observations of fine and coarse mode AOD sampled based on the Moderate Resolution Imaging Spectrometer (MODIS). Khade et al. (2013) explored the possibility to estimate soil erodibility factors (that drive dust emissions) by assimilating satellite AOD in an ensemble adjustment Kalman filter. Xu et al. (2017) showed the usefulness of assimilating both reflected solar and infrared radiances from the CLARREO's mission to constrain accurately size resolved aerosol emissions for four dust size bins. Further, they concluded that CLARREO data failed to constrain dust sources due to its narrow swath, and the combination of narrow and wide swath observations might be more desirable. Possibly the full PACE mission observations, that includes a narrow (SPEXone) and a wide (HARP-2) swath polarimeter, as well as a wide swath radiometer (OCI), would be able to bring this idea into practice.

In this study we quantify how well an instrument with high accuracy but limited coverage, like SPEXone, can estimate aerosol emissions. Under the framework of OSSEs, we implement an existing Local Ensemble Transform Kalman Smoother (LETKS) code to operate with the global aerosol climate model ECHAM-HAM and assimilate synthetic observations based on a future multi-angle polarimeter instrument (SPEXone) and a theoretical satellite with near perfect global coverage. Following the results of our previous work and based on the MAP observational capabilities of SPEXone, we assimilate $AOD_{550}$, $AE_{550-865}$ and $SSA_{550}$ in order to encompass information related to aerosol mass, size and absorption (Tsikerdekis et al., 2021). In Section 2, the SPEXone instrument on PACE and the aerosol climate model ECHAM-HAM are described, along with the spatiotemporal coverage and uncertainties of SPEXone and of an idealized instrument. Section 3, presents the data assimilation system, its newly developed features and the experimental set up. Finally, in Section 4, the ability of SPEXone to estimate emissions is presented accompanied with SPEXone sensitivity experiments and other sensitivity experiments that explore uncertainty factors that can affect the emission estimation.

## 2 Data

### 2.1 SPEXone on PACE

SPEXone is a passive remote sensing MAP instrument, part of the NASA Plankton, Aerosol, Cloud, and ocean Ecosystem
(PACE) mission (Werdell et al., 2019), scheduled for launch in 2023/2024. It was developed by the Netherlands Institute for
Space Research (SRON) and the Airbus Defense and Space Netherlands (ADS-NL) with optical expertise from the Netherlands
Organization for Applied Scientific Research (TNO). SPEXone can measure intensity and polarization of backscattered sun
light, at multiple wavelengths and discrete viewing angles for a specific pixel on the ground. Specifically, it can measure
radiance and polarization at 5 viewing angles (+57°, +20°, 0°, -20°, -57° on ground) with high accuracy (0.003) on the Degree
of Linear Polarization (DoLP). SPEXone is a spectrometer, measuring a continuous spectrum (at 2 nm resolution for radiance
and 10-25 nm for polarization) between the spectral range from 385nm to 770nm. Sensor's horizontal resolution is ~5.4 x
4.6km for all viewing angles and the swath is 100km. The aerosol retrieved parameters include column AOD, AE, SSA, aerosol
layer height, effective radius, effective variance (of the size distribution), complex refractive index, particle number for a fine
and a coarse aerosol mode, in addition to a shape parameter for the coarse mode. Detailed information on the optical and
technical attributes as well as the retrieval capabilities of SPEXone can be found in Hasekamp et al. (2019) and van Amerongen
et al. (2019).

### 2.2 The ECHAM6-HAM2 Aerosol Climate Model

The sixth generation of the general circulation model ECHAM6, developed at the Max Planck Institute for Meteorology (MPI-
M) in Hamburg, Germany (Stevens et al., 2013) and the second version of the Hamburg Aerosol Model (HAM2) (Stier et al.,
2005; Tegen et al., 2019; Zhang et al., 2012) are used to simulate the physical and chemical processes of aerosol in the
atmosphere.

The M7 aerosol module used in HAM2 considers five aerosol species, dust (DU), sea salt (SS), organic carbon (OC), black
carbon (BC) and sulfates ($SO_4$) (Vignati et al., 2004). Aerosols are partitioned in seven unimodal lognormal particle size
distributions (Nucleation, Aitken, Accumulation, Coarse) called modes, separated into two hygroscopic classes (hydrophobic
and hydrophilic). Six of these modes contain several aerosol species (internally mixed modes). Each mode is characterized by
the number concentration and the mass concentration by species. Aerosol number and mass are used in order to calculate the
median radius for each mode (Tegen et al., 2019). The mode width (standard deviation of the lognormal distribution) is
assumed and fixed, equal to 1.59 for nucleation, Aitken and accumulation and 2.00 for the coarse mode. The cloud and aerosol
optical properties are computed using Mie Theory and derived from lookup tables (Tegen et al., 2019) using the prognostic
concentrations of aerosol tracers (Schultz et al., 2018).

All aerosol species are emitted, transported, deposited and take part in aerosol-radiation interactions (scattering and absorption)
as well as aerosol microphysical processes (e.g. nucleation, coagulation, aerosol water uptake and cloud activation). The
natural aerosol types (DU, SS) are introduced to the atmosphere by utilizing the simulated information of wind and certain

surface and ocean characteristics. Other aerosol species (OC, BC) or aerosol precursor gases ($SO_2$, DMS) that are emitted from

both natural (e.g. forest fires) and anthropogenic sources use predefined emission inventories (Zhang et al., 2012). For a description of the importance of individual processes, see the budget by species in Schutgens and Stier (2014).

Two SS emission schemes are used in this study. The first and default scheme in ECHAM-HAM, parameterizes sea salt emissions based on laboratory measurements (Keene et al., 2007) using the wind velocity at 10m and the sea surface temperature (SST) (Long et al., 2011; Sofiev et al., 2011). Low SST results in lower sea salt emissions with smaller particle

size (Sofiev et al., 2011). The second scheme (previously the default option) in ECHAM-HAM, calculates the sea salt flux mass and number through tables of wind speed classes and fits to two lognormal distributions based on Guelle et al. (2001 and reference therein). Note that sea salt particles are emitted only in the soluble accumulation and coarse mode in both schemes. Dust emissions are based on the dust source scheme developed by Tegen et al. (2002). Wind velocity at 10m is the main driver of dust aerosol particle production while soil properties are also taken into account. The preferential dust emission sources are

consisted by arid or low vegetated areas and are predefined based on Tegen et al. (2002). Improvements on the surface aerodynamic roughness length, soil moisture and soil properties specifically over East Asia were made by Cheng et al. (2008). The threshold friction velocity depends on the soil size distribution, vegetation cover and soil moisture (Cheng et al., 2008). Further, updates related to the representation of Saharan dust sources were made using infrared dust index from the SEVIRI instrument upon Meteosat Second Generation Satellite by Heinold et al. (2016).

The emission for the remaining aerosol types and aerosol precursors are defined using emission inventories derived for 14 sectors. Each sector may include one or more aerosol types or aerosol precursors (Schultz et al., 2018; Tegen et al., 2019). The Atmospheric Chemistry and Climate Model Intercomparison Project (ACCMIP) dataset is used for the anthropogenic, biomass burning and aerosol precursor emissions, consisting of monthly mean estimates at a horizontal resolution 0.5°x0.5° (Lamarque et al., 2010). The Community Emissions Data System (CEDS) is used as an alternative for the anthropogenic aerosol and

aerosol precursor (Hoesly et al., 2018). The first version of Global Fire Assimilation System (GFAS) is also used for the biomass burning emissions coming from grass and forest fires consisting of daily gridded estimates at 0.5°x0.5° horizontal resolution based on the fire radiative power measurements of MODIS instrument (Kaiser et al., 2012). A more detailed description of both ECHAM6 and HAM2 can be found in Tegen et al. (2019).

# 3 Methods

## 3.1 Local Ensemble Transform Kalman Smoother

The Local Ensemble Transform Kalman Smoother (LETKS) is used to estimate aerosol emission fluxes. This method has been previously used by Schutgens et al. (2012) for aerosols emission estimation and earlier by Bruhwiler et al. (2005), Peters et al. (2005) and Feng et al. (2009) for $CO_2$ emission estimation. It requires a model to produce background information based on assumed emissions and observations that are assimilated to estimate analysis emissions. In data assimilation studies the terms

analysis or posterior are used to describe the improved state of the system due to assimilation, although in this study we reserve

the term analysis for cases where the aerosol emissions were estimated by a fraction of the total observations that are going to affect them in the end (more details follow).

The data assimilation occurs in Assimilation Cycles, where each cycle contains a background and an analysis step as depicted in Figure 1. Dashed boxes indicate the default emission where no assimilation took place yet, while filled boxes indicate emission changed based on observations. The background step consists of an 8-day ($\Delta T_b$) forward simulation of the model that initially (1$^{st}$ cycle) will create the simulated background observations. Next all the available observations within the last 2 days ($\Delta T_s$) of the forward simulation are assimilated in order to estimate the analysis emission for the last 6 days ($\Delta T_a = \Delta T_b - \Delta T_s$) of the forward simulation. Note here that the term analysis is used to indicate the updated emissions affected by n days of observations (where n < $\Delta T_a$), while the term posterior is used to indicate updated emissions affected by $\Delta T_a$ days of observations (Figure 1). This is where the 1$^{st}$ cycle ends. In the 2$^{nd}$ cycle background emissions are set equal to the analysis emissions of the 1$^{st}$ cycle. Then the respective steps of the background and assimilation steps are performed for the 2$^{nd}$ cycle. This process continues until the end of the assimilation experiment.

The assimilation window ($\Delta T_s$), defines the shift (step) in time of the forward simulation in each cycle, the period of the assimilated observations and the period during which emissions are estimated. A $\Delta T_s$=2days allows the aggregation of more satellite observations that provides a better constrain for emission estimates globally, but it also assumes that emissions do not change considerably over this period. Undoubtedly this is not always the case, for example dust emissions may vary a lot from day to day. The smoother lag ($\Delta T_a$) determines how many days are going to be affected by the assimilated observations in one assimilation cycle. In our setup this is equal to six days, but we conduct experiments to see its impact when reduced to four and two days.

Note that it is assumed that the observations of a certain day contain only a fraction of the available information to change the emissions and the rest is contained in observations of subsequent days. Thus, emissions should be estimated iteratively, allowing observations up to 6 days after to correct the emissions. The posterior emission perturbations, corrected by 6 days of observations, are derived after 3 assimilation cycles and are indicated with an asterisk (*) in Figure 1. For example, the posterior emission perturbations for the days 7 and 8 are estimated in the 3$^{rd}$ assimilation cycle and are corrected from the assimilated observations of days 7 to 12.

Background emissions come with uncertainties. The uncertainty of background emissions are represented by an ensemble that is generated by perturbing the default emissions. The perturbations are not globally constant but vary from grid cell to grid cell. Each grid cell has a distinct prior emission distribution. Changes in neighboring grid cells of each member are not abrupt but smooth, This spatial correlation of the prior perturbations was generated using spatial smoothing, a method where data points are averaged with their neighbours. A step by step description on how our spatially correlated perturbations are created can be found at subsection 3.2 of our preceding work (Tsikerdekis et al. 2021). The spatial correlation length scale of the generated perturbations is approximately 25° omnidirectionally. The perturbations are uniquely created and distinctively estimated by the data assimilation system for each aerosol species and sulfate precursor gas. The resulting 2D spatially correlated perturbations are multiplied with the model's emissions for each aerosol species and each member, resulting in an

ensemble of simulations. In our experiments the ensemble size is 32. Note that the mean and the standard deviation of background distribution is equal to 1. Furthermore, it is noted that the perturbations are uniquely defined every $\Delta T_s$=2days (different colors in the boxes of Figure 1). The rationale here is that the simulated observations and emissions at day D (where D is any integer number) will be more correlated than the simulated observations at day D+$\Delta T_s$ and emissions at day D. Consequently, changes in emissions caused by assimilated observations of day D will be stronger compare to changes in

emissions by assimilated observations of day D+$\Delta T_s$. This design is based on the fact that observations on the day of the emissions carry more information about the emissions, than observations in subsequent days.

More info regarding the emission perturbations and the ensemble can be found in our preceding work (Tsikerdekis et al., 2021). New emission estimates are obtained by estimating new perturbed emission factors based on the assimilated observations by solving the Kalman filter equations:

$$x_a = x_b + P_a \cdot H^T \cdot R^{-1} \cdot (y - H \cdot x_b) \tag{1}$$
$$P_a = (I + P_b \cdot H^T \cdot R^{-1} \cdot H)^{-1} \cdot P_b \tag{2}$$

where $\mathbf{x_b}$ is the background state vector and represents the variables aimed to be improved. It includes emission perturbations of five aerosol species (DU, SS, OC, BC, SO$_4$) along with the emission perturbations of two aerosol precursor gases (SO$_2$, DMS). $\mathbf{x_a}$ is the analysis state vector, the improved version of $\mathbf{x_b}$ based on the assimilated observations ($\mathbf{y}$). The background

and analysis uncertainty and correlations of emission are represented by the model error covariance matrix $\mathbf{P_b}$ and $\mathbf{P_a}$ respectively, using the ensemble. The observational uncertainties are represented by the error covariance matrix $\mathbf{R}$. We assume R to be diagonal (i.e. correlations between observational errors are assumed to be zero always). The observational operator $\mathbf{H}$, translates the emission perturbations ($\mathbf{x}$) to the simulated observations ($\mathbf{H \cdot x}$) and it is entirely handled by the model (emission, transport, deposition, aerosol processes and optical properties code). T stands for the matrix transpose operator.

**3.2 LETKF-Smoother Prior Correction**

The ensemble Kalman filter assumes that prior emissions in the model are unbiased. In reality this is not necessarily the case, since emission inventories or emission schemes in models may suffer from biases that are often higher than the defined background uncertainty. Past studies have demonstrated that optimizing prior emissions based on previous assimilation cycles can improve data assimilation performance (Bruhwiler et al., 2005; Peng et al., 2017; Peters et al., 2005). Based on that we

have developed a method, hereafter called the "prior correction". Prior correction updates the prior emission based on estimated emissions from the previous assimilation cycles, thus correcting biased emissions of the model as the data assimilation experiment progresses in time. Specifically, the ensemble mean (E$_{mean}$) of the new emission perturbations of each cycle is defined according to the analysis results of the previous assimilation cycle. For example, the E$_{mean}$ of the newly created perturbations at B (Figure 1) will be equal to the E$_{mean}$ of the perturbation at A (Figure 1). Consequently the filter corrects the

emissions bias based on the estimated emissions of previous assimilation cycles.

Although prior correction fixes the problem of potentially biased model prior emissions, it may introduce unwanted negative emission perturbations when the E$_{mean}$ drops below 1. One way of addressing this issue would be to set all negative produced

perturbations to zero, but this will affect the distribution of the perturbations and make it less gaussian. Hence, the ensemble standard deviation ($E_{std}$) is adjusted according to the $E_{mean}$:

$$E_{mean} \leq 1.1 \rightarrow E_{std} = 1 \tag{3}$$

$$E_{mean} > 1.1 \rightarrow E_{std} = E_{mean} \cdot 0.9 \tag{4}$$

As an example, three distributions with different $E_{mean}$ and adjusted $E_{std}$ are depicted in Figure 2. Note that even under this design there is <0.5% chance to generate a negative value in the distribution when $E_{mean}$ is lower than 1, which in that case is set to zero. The adjusted $E_{std}$ method implies that emissions will have lower relative background uncertainty when $E_{std} < 1.1$.

This might not benefit the data assimilation system for some dust sources where emissions from day to day can differ substantially, although we haven't notice examples where this is a problem.

The prior correction approach has two optional settings where the background $E_{mean}$ can reach a maximum or a minimum threshold. Under the framework of OSSEs, these background minimum and maximum values are known, since the background and the nature emissions can be compared. However, in reality these values can only be approximated using observations, for

example, by the ratio of the background simulated observations to real observations. The majority of the experiments with the prior correction option use a minimum and a maximum threshold equal to 0.3 and 3.6 respectively, based on the AOD ratio of NAT to CTL. It is noted however that AOD is just one of the assimilated observations that constrains the emissions and further work is needed in case background minimum and maximum settings are used in a data assimilation experiment with real observations. The effect of prior correction is tested by conducting two data assimilation experiments (with and without prior

correction) that are presented in Appendix A.

### 3.3 Observing System Simulation Experiments (OSSEs)

Observing System Simulation Experiments (OSSEs) are data assimilation experiments in which synthetic observations are used that themselves are generated by a model. The synthetic observations of an OSSE can be modified to match the spatiotemporal coverage and observational uncertainty of any satellite sensor. Hence with OSSEs it is possible to assess the

potential impact of past, present and future satellite missions on aerosol top-down emission estimation. The unique advantage of OSSEs is that the "truth" is perfectly known for all times, locations and climate/aerosol components and can be used to evaluate the performance of an experiment.

There are three parts of an OSSE, (i) the Nature Run (NAT) that represents the "true" conditions of the aerosol state in the atmosphere, (ii) the Control (CTL) run of the model, which sets the baseline performance of the model without data being

assimilated, (iii) and the data assimilation run (DAS) where synthetic observations are assimilated in a model identical to the CTL model in order to improve aerosol emissions. The intercomparison between the differences CTL – NAT and DAS – NAT can provide the added value of the assimilated observations, identify limitations of the data assimilation system or quantify the role of some processes on the estimated emissions. The main goal of the present paper is to assess the ability of different satellite observations on quantifying aerosol emissions. Therefore, for all experiments we use the same physical model for the

NAT, CTL and DAS, because otherwise we cannot attribute differences between NAT and DAS to either limitations of the

satellite observations or model differences. We also perform some additional experiments with different nature runs (NAT_M, NAT_E) to assess different causes of uncertainty in emission estimation (e.g. biased meteorology), in addition to the standard nature run (NAT), and partially address the OSSE identical twin problem (Arnold and Dey, 1986; Timmermans et al., 2015). Note that the meteorology of all experiments is nudged to the ensemble mean of the 10 analysis member of ERA5 (Hersbach et al., 2020), except the nature run NAT_M (details below).

The standard nature run (NAT) changes only the emissions in comparison to CTL, by multiplying the default emissions of DU and SS with 0.5, the default emissions of OC and BC with 2 and the default emission of $SO_4$, $SO_2$ and DMS with 1.5. These emission factors are within the current range of uncertainty of aerosol emissions (discussed in Tsikerdekis et al. (2021)) and create a distinct difference in the global and regional distribution of AOD, AE and SSA in comparison to CTL. These emission factors are chosen arbitrary, aiming to test if the data assimilation is able to estimate them correctly (test the system), rather than to reduce biases between NAT and a specific set of observations of an existing satellite (e.g POLDER-3). Nevertheless the differences between CTL – POLDER and CTL – NAT exhibit similarities in the biomass burning region in the Tropics and the global ME and MAE of these differences are on the same scale (not shown). The second nature run (NAT_M) uses the same altered emissions as NAT but its meteorology is nudged to reanalysis ERA-interim. Consequently, the assimilated observations sampled from NAT_M can show the impact of biased meteorology on emission estimation. To investigate whether the scaling of emissions in NAT represents a too simple difference between nature and data assimilation run, a new nature run (NAT_E) was performed that changes emission parameterizations schemes for DU and SS, and uses different emission inventories for the other species. This approach creates distinct spatiotemporal differences between the two runs in each species. An overview of all the NAT emission choices are depicted in Table 1.

## 3.4 Instruments coverage and uncertainty

The SPEXone spatial coverage at native resolution (~5.4 x 4.6km) was simulated using an orbit simulator for cloud-free pixels based on the MODIS cloud product. In our case, we would like for SPEXone spatial coverage to be consistent with ECHAM clouds, thus we modified the SPEXone spatial coverage to match ECHAM cloud-mask. The goal of this post-process was to create an ECHAM cloud-based SPEXone mask that provided similar amount of observations as with the MODIS cloud-based SPEXone mask (more detail at Appendix B).

An ideal sensor in terms of spatial coverage was assumed in order to test the data assimilation system and act as a benchmark for the SPEXone ability to estimate aerosol emissions. This sensor, hereafter mentioned as SUPER, is able to retrieve $AOD_{550}$, $AE_{550-865}$ and $SSA_{550}$ over the whole globe every 2 days. The 2-day global coverage was based on the step of the data assimilation set which estimates the emissions every two days. In addition, SUPER sensor is able to get aerosol observations even over cloudy pixels and over very high latitudes.

The spatial coverage for a two day period for these two satellites is shown in Figure 3. Note that SUPER has a fixed number of observations in time and space while the number of SPEXone observations fluctuates in time and space depending on cloud cover and orbit characteristics. The total number of grid cell observations (each grid cell includes an $AOD_{550}$, $AE_{550-865}$ and

SSA$_{550}$) assimilated for the period 20$^{th}$ of July to 20$^{th}$ of September of 2006, is more than twice in SUPER (139872) compared to SPEXone (61086). The observations we are using are super observations, meaning that all the high resolution SPEXone observations were aggregated to the model resolution (1.875° x 1.875°). At the original resolution of SPEXone, our SUPER sensor would provide approximately 6 times more observations. Note that in that case these observations would be very closed together and highly spatially correlated. Also, with super observations the swath of SPEXone appears larger than 100 km, since only one high resolution SPEXone resolution within each grid box is needed to provide a value for the whole grid box of a size 1.875° x 1.875° (~250 km).

An instrument/retrieval simulator was used to generate estimates of observational errors. Retrievals for four individual days were used for that. Specifically the estimated uncertainty is based on the difference between the retrieved and the true values, following a similar method as Tsikerdekis et al. (2021). More details at Appendix C. Note that these observational uncertainties were used for both satellites (SUPER and SPEXone).

## 3.5 Experimental Setup

All the experiments span 2 months in the summer of 2006 (20/07/2006 to 20/09/2006). The year and period was chosen based on our previous work (Tsikerdekis et al., 2021). Prior to this period the model span up for 3 months (01/04/2006 to 01/07/2006) and the ensemble background emissions span up for 20 days (01/07/2006 to 20/07/2006). We employ a grid resolution T63L31 (1.875° x 1.875°, with 31 hybrid-sigma vertical layers concentrated in the Troposphere).

There are a few LETKF parameters that can be adjusted. In this study we keep these parameters fixed in all of our experiments. The description, discussion and sensitivity experiments of these parameters (ensemble size, inflation local patch size and the horizontal localization) was presented in our preceding study (Tsikerdekis et al., 2021). The data assimilation ensemble size consists of 32 members. The local patch size and the horizontal localization are set to 8 and 4 grid cells respectively, while the inflation is set to 1. The inflation parameter is essentially deactivated with the value equal to 1, since under the emissions estimation setup of the data assimilation system the background uncertainty remains big enough throughout the experiment for the data assimilation to work. The local patch size is deliberately chosen to be high (8), in order to let observations that are far away from the source (up to 15°) impact the emission estimation.

Table 2 shows the list of experiments related to SPEXone. The experiment where the assimilated observations are based on the SUPER spatiotemporal sampling is used mainly as a benchmark for the performance of the experiments that use the SPEXone sampling. The experiments where the assimilated observations use the SPEXone satellite coverage intend to evaluate the added value of SPEXone ability to estimate emissions under different observational uncertainty and data assimilation options. Specifically, SPX used the default errors estimated for SPEXone retrievals (Appendix C). The experiment SPX_2U doubles the uncertainty of the assimilated observations and SPX_2URE doubles the uncertainty and adds random errors (with standard deviation equal to the observational uncertainty) to the assimilated observations. Finally SPX_W1 and SPX_W2 reduce the $\Delta T_a$ length to 4 and 2 days respectively (from 6 days originally), hence less observations are used to derive the analysis emissions in each assimilation cycle and only 2 and 1 assimilation cycles (instead of 3) are used to calculate the

analysis emission perturbations. Consequently, the data assimilation experiment is faster and less computationally expensive, but less observations used to obtain the analysis emission.

Sensor SUPER is further used in other sensitivity experiments that aim to assess issues related to the nature run complexity and development of the data assimilation system (Table 3). The SUP0_M experiment points out the degradation in emission estimation purely due to biased wind by assimilating observation from NAT_M. SUP_E assimilates observation from NAT_E and shows that even under totally different emission schemes and emission inventories between the nature run and the data assimilation experiment, the emission errors are reduced.

### 3.6 Data assimilation initialization

The prior emissions may be over or under estimated and the smoother (+ prior correction) will take time to adjust them. The smoother's time-window of 6 days suggests that correct estimation of emissions does not happen until a multiple of that number of days has passed. During this period, the smoother is adjusting to the major biases present in the CTL emissions. Based on the results of our data assimilation experiment we define this period, in order to exclude it from the evaluation that follow in the results section.

Figure 4 shows that the differences between DAS and NAT (solid lines) reach a value close to zero after 26 days. From that point on till the end of the experiment, these differences fluctuate around zero. For comparison the emission differences of CTL – NAT (dashed lines) are also shown. Note that the day-to-day dust and sea salt emissions differences can fluctuate a lot in CTL, but SUP is able to estimate them adequately.

The duration of the initialization phase may be expected to be a multiple of the longest of two time-scales: the aerosol life-time (that determines how quickly aerosol are deposited) and the DA window (that determines how quickly we can adjust emissions based on observations).

This is shown in Figure 5, where approximately after 26 days the aerosol optical properties differences and column burden relative differences between DAS and NAT reach a value close to zero and start fluctuating around it till the end of the assimilation experiment. Consequently, we choose the period of 26 days as the data assimilation initialization period and only the remaining 36 days are evaluated in the result section, spanning from 2006/08/15 to 2006/09/20. Note that data assimilation initialization varies for each experiment, depending on the amount of the assimilated observations, the differences with nature run as well as several assimilation options. Nevertheless, 26 days are sufficient as a data assimilation initialization period for all experiments (not shown), except SUP_E for SS emissions in the coarse mode, thus it is kept constant throughout the paper.

## 4 Results

### 4.1 Emission estimation using SPEXone

The ability to estimate the true aerosol state using SPEXone are compared to an experiment in which observations were assimilated based on a sensor like SPEXone (meaning that it can retrieve the same type of observations with the same accuracy)

but with an almost perfect global coverage. In order to understand the simulated aerosol state for the examined period, the aerosol optical properties of the CTL experiment are shown and discussed in Figure 6. High AOD is evident over Sahara and

Arabian Peninsula mainly due to dust, over the tropical forests (Amazon, Africa, Indonesia) mainly due to organic and black carbon and over Europe, North America and China mainly due to sulfates. AE is small over isolated ocean areas which are dominated by sea salt and high values over land, excluding desert areas where large dust particles prevail. High AAOD (low SSA) highlights high black carbon concentrations, either from natural (biomass burning) or anthropogenic (fossil fuel) sources and intermediate values over high sources of dust. Note that SSA (not AAOD) is the quantity that is assimilated in our system

(for details on differences between SSA and AAOD assimilation see Tsikerdekis et al. 2021), but AAOD is shown in the plots since it is easier to interpret.

The ability SPEXone and SUPER sensors to recreate the NAT are summarized in Figure 7, where the differences between the experiments CTL, SUP and SPX from NAT are depicted for AOD, AE and AAOD. In both data assimilation experiments the modelled aerosol are improved when compared to the CTL experiment and the global Mean Error (ME) as well as the global

Mean Absolute Error (MAE) is almost zero. ME and MAE equations can be found in Appendix B of our preceding publication (Tsikerdekis et al., 2021). The performance of SPX is as good as the SUP, which suggests that the spatial coverage of SPEXone is sufficient to constrain the emissions in a similar fashion as the SUPER satellite.

An important advantage of OSSEs is that we are able to evaluate the estimated emissions of the data assimilation experiments with the emissions of the nature run. Figure 8 and Figure 9 depicts the emission of aerosol species for NAT and the emission

differences for CTL, SUP and SPX from NAT. In both data assimilation experiments the estimated emissions are improved compared to the emission of the CTL. The overestimated dust emission in the CTL are constrained in the data assimilation experiments and only in the western part of the Sahara desert where emission are high, the ME is not close to zero. For both data assimilation experiments the relative ME averaged for the same region is lower than 10% (not shown). The overestimated sea salt emissions in CTL are constrained globally in both data assimilation experiments, though in SPX the sea salt emission

over the Indian Ocean shows high ME with relative ME in some grid cells that exceeds 50%. This is caused by the limited observations by SPEXone due to cloudiness over India and surrounding seas (see FigureA 3). The ME and the relative ME emission for organic and black carbon over high sources, mainly over the Tropics in South America, Africa and Indonesia but as well as eastern China, reach almost zero in the data assimilation experiments. Sulfates in the model are mainly produced from $SO_2$ precursor emissions and only a small fraction (2.5%) of sulfates are directly emitted to the atmosphere. For all other

species (DU, SS, OC and BC) the assimilated aerosol optical and microphysical observations directly constrain the emission of the particles that form these observations in the atmosphere. Despite that, the $SO_2 + SO_4$ emissions are constrained reasonably well and especially over high anthropogenic sources (North America, Europe, India and China) the relative ME by grid cell does not exceed 10% (not shown) in both data assimilation experiments. These results suggest that SPEXone limited observational coverage can estimate global aerosol emission in a similar manner as a sensor that would have an almost perfect

observational coverage. Although it is noted that local error due to cloudiness deteriorate the performance of SPEXone in

comparison to SUPER. Further, we assume that 1.875 degree aggregate of SPEXone contain non-significant representation error and the observations of both sensors are unbiased.

## 4.2 Emission estimation using SPEXone – Sensitivity experiments

A series of data assimilation experiments were conducted in order to explore a less optimistic (SPX_2U) scenarios for the
SPEXone retrievals, and also one to check what the effect is of adding actual noise to the observations  (SPX_2URE) instead of relying purely on the uncertainty descriptions of the measurements.  Further, we vary  the $\Delta T_a$ length (SPX_W1, SPX_W2) of the data assimilation system. The differences of these two data assimilation experiments from NAT for AOD, AE and AAOD are depicted in Figure 10. In all cases the observations improve compared to the CTL experiment (Figure 7 a-c), although not to the extent of the default experiment SPX which was discussed in the previous subsection (Figure 7 g-i).
Specifically, SPX_2U, where the assimilated observations uncertainty was doubled, shows similar results for AOD and AE while the AAOD bias is increased slightly in comparison to SPX (Figure 10 a-c). SPX_2URE, where the assimilated observations uncertainty was doubled and random errors (with standard deviation equal to the observational uncertainty) were added to the assimilated observations, the bias increases over northeastern China for AOD, over Sahara, Arabian Peninsula and North Indian ocean for AE and over Tropical Africa and the Amazon basin for AAOD (Figure 10 d-f). We can quantify
the effect of observations random errors on emission estimation by comparing the experiments SPX_2U and SPX_2URE. The data assimilation performance does not degrade significantly when taking into account random errors in the assimilated observations. Specifically the dust emission global MAE increases by 5 percent points due to random errors, while for other species the increase is even lower (Figure 13).

SPX_W1 and SPX_W2 reduce the $\Delta T_a$ length to 4 and 2 days (from 6), hence less observations are used to derive the analysis
emissions in each assimilation cycle and only 2 and 1 assimilation cycles (instead of 3) are used to calculate the analysis emission perturbations. The results reveal that $\Delta T_a$=4 days (SPX_W1) is sufficient to constrain the AOD, AE and AAOD in a similar manner as a  $\Delta T_a$=6 days (SPX) (Figure 11 a,b,c). In other words, under the current experimental setup, observations 5 to 6 days after the emissions probably hold very little information for the correction of these emissions, and their exclusion has a very limited impact on the data assimilation performance. Contrary the experiment SPX_W2 shows a degradation in
performance over western Sahara and North Atlantic for AOD and AE (Figure 11 d,e,f), indicating that observation in subsequent days 3 and 4 hold useful information for the correct estimation of emissions at day 1 and 2 as discussed in the next paragraphs. Note that SPX_W1 and SPX_W2 need ~33% and ~66% less computational resources than SPX respectively, since the background step in each assimilation cycle is shorter.

Figure 12 shows the mean and standard deviation of errors per grid cell. These errors are averages for the evaluation period of
the difference between an experiment (CTL or DAS) and NAT. Both SUP and SPX errors are significantly smaller than CTL in both global (mean) and local errors (spread). The global AOD MAE of SPX_2U and SPEX_2URE remains very low, while AE and AAOD global ME slightly increase. Note that SPEXone AOD uncertainty range (Appendix C) is very low (lower than 10% over ocean and on average 15% over land), and doubling this uncertainty has only a limited effect on the analysis. On the

other hand, the uncertainty of AE and SSA observations is higher than AOD, hence the data assimilation performance is
affected to a larger extent. Overall, it can be concluded that in these less optimistic (doubled uncertainty), the assimilated observations based on SPEXone spatial coverage are still able to estimate the emissions with reasonable accuracy. Further, the experiment where actual noise is added to the measurements shows similar results to the experiment where no noise was added. This illustrates that the system is not 'overfitting' the observations but takes the specified uncertainty correctly into account even when there is no noise added to the measurements.

In terms of estimated emissions, the four sensitivity experiments rank a bit lower in comparison to both SUP and SPX as indicated in Figure 13, where the global relative MAE for various species is shown. Specifically, SPX has similar emission errors to SUP, but differ in the SS estimated emission which is caused by the limited observations in SPEXone due to cloudiness over India and surrounding seas (see FigureA 3), as discussed in the previous subsection. SPX_2U and SPX_2URE emission biases for all species are increased no more than 10 percent points, in comparison to SPX, which indicates that
increased (double) uncertainty and adding random errors in the observations has a noticeable but small negative effect on the global relative differences in the emissions. Finally, SPX_W1 emission bias increases no more than 6 percent points in comparison to SPX in all species. However, dust emission error grows to 54% in SPX_W2 from 17% in SPX_W1, indicating that the information content of observations 3 and 4 days after the emissions is very rich and it should be used to correct these emissions, especially for Sahara dust plumes that extent over the Atlantic Ocean and last for several days. The emissions of
OC, BC and $SO_2+SO_4$ are estimated very accurately by all of the data assimilation experiments, with relative MAE ranging from 0% to 5%, which indicates that in terms of the global mean emission estimation these emissions are unaffected by the sensor spatial coverage and observational uncertainty increase that were tested. The global maps of emission differences from NAT for the four sensitivity experiments of these subsection are shown in FigureS 1.

## 4.3 Other sources of uncertainty for emission estimation

OSSEs also allow us to quantify the uncertainty due to assumptions in nudging meteorology or emission source locations. The first relates to the assumption that the meteorological part of the model and specifically the wind components (U & V) are perfect. The second factor relates to complex spatiotemporal change of aerosol emission in the nature run compare to the data assimilation run and test if the system is able to estimate the correct emissions when the data assimilation and nature runs emissions differ by not just an emission factor (per species) which is constant in time and space.

### 4.3.1 The effect of biased meteorology

The OSSEs in previous subsections implicitly assumed that the data assimilation experiment would have perfect knowledge of the NAT meteorology. Since even reanalysis datasets of windspeeds have errors, we here test their impact. Simulations that were nudged to different reanalysis datasets (e.g. ERA-interim and ERA-5) reveal very dissimilar results in terms of AOD, AE and SSA for specific regions (Figure 14 g,h,i).

In this subsection we explore the effect of biased meteorology in the aerosol emission estimation by nudging the wind components of nature run (NAT_M) to ERA-interim and the wind components of the data assimilation (SUP0_M) experiment to ERA-5. The sampled observations of NAT_M are based on the SUPER sensor; hence the observational coverage is optimal in space and continuous in time. Note that the emissions of NAT_M are scaled with the same scale factor as NAT (Table 1). Further, prior correction is not used in SUP0_M.

The evaluation of SUP0_M modelled aerosol against NAT_M reveal high errors in some regions (Figure 14 d,e,f). Unsurprisingly, AOD differences of SUP0_M – NAT_M with NAT – NAT_M in Figure 14 display striking similarities on subtropical and tropical African continent and Atlantic Ocean, as well as over East China Sea and Philippine Sea, which suggests that the remaining aerosol biases on SUP0_M are mostly related to the biased meteorology that affects aerosol transport paths.

In terms of the estimated emissions, SS are negatively affected the most by the effect of biased meteorology. Figure 15 shows that the relative MAE in SS emissions increases by 24 percent points in SUP0_M (42%) compared to SUP0 (18%) while the estimated emissions of DU, OC, BC and $SO_4+SO_2$ are negatively affected by the effect of biased meteorology to a smaller extent (~10%). Also, the comparison of the two grey bars, CTL (NAT) and CTL (NAT_M) shows that the different meteorology changes significantly the DU emissions and to a lesser extent the SS emissions. Note that regional error (estimated

for each grid cell) can be higher than what is indicated in Figure 15. The global map emissions differences for CTL – NAT_M, SUP0_M – NAT_M and NAT – NAT_M are shown in (FigureS 2).

Transport deviations (vertically and horizontally) between ERA-5 and ERA-interim were assessed using Lagrangian transport simulations by Hoffmann et al. (2019). In that study differences of Lagrangian simulations based on the two reanalysis products were up to 2 to 3 orders of magnitude compared to differences caused by parameterized diffusion and subgrid-scale wind

fluctuation after 10 days. Some of the main simulation improvements of ERA-5 compared to ERA-interim are its higher spatial (31km) and temporal (hourly analysis) resolution as well as it's 4D-var uncertainty estimate, which comes from a 10-member ensemble of data assimilation in a coarser resolution (63km). Considering the improvements of ERA-5 compared to its predecessor, we assume that the aerosol differences (Figure 14 g,h,i) caused by nudging ECHAM-HAM to ERA-5 or ERA-interim represent a worst case scenario and the differences between ERA-5 and the real wind are not greater than that.

**4.3.2 The effect of using different emission inventories between nature and data assimilation runs**

Our nature run (NAT) has emissions that are simply scaled for the different species compared to the control and data assimilation runs. To investigate whether this scaling represents a too simple difference between nature and data assimilation run, we conduct OSSEs with a new nature run (NAT_E). In this new nature run we change the emission inventories and emission schemes (Table 1) compared to the control and data assimilation runs. This creates a more realistic emission

differences between NAT_E and CTL that fluctuate in time and space. CTL – NAT_E differences in Figure 16 illustrate an overestimation of AOD and AAOD over the tropics in South America and Africa. An underestimation of AOD is apparent in

Southeast Asia and over the deserts in western Sahara and Taklamakan. In addition, a strong global overestimation (0.46) of AE, mainly over the ocean, is observed due to a high amount of SS coarse particles emitted by the scheme selected in NAT_E. In a new assimilation experiment (SUP_E) we used some new options. Emission estimation was conducted by mode and not only by species (separately for accumulation and coarse) for the SS and DU aerosol species. In addition, prior correction was used (without the prior max option). Both of these changes were introduced for the SUP_E experiment in order to create more variation in AE and let emissions of SS in the coarse mode match those in NAT_E, which are much higher than the background uncertainty for mid and high latitudes. Results of data assimilation experiments, where we applied these two changes one at a time, are shown in FigureS 3.

In SUP_E, we perform a data assimilation experiment using the CTL baseline prior emissions with observations drawn from NAT_E. The data assimilation system was able to adjust model emissions in order to match the observations of NAT_E. Specifically the global ME for SUP_E is zero for AOD and AAOD, while AE global ME is reduced from 0.46 to 0.11 (Figure 16), with the highest local errors still persisting over high latitudes (FigureS 4 and explanation in caption).

The global relative MAEs for emissions are depicted by species in Figure 17 for SUP_E and CTL. The emission errors of SUP_E for all species are reduced or remain almost unchanged ($SO_4+SO_2$) compare to CTL. Although NAT_E uses very different emission inventories compare to SUP_E, the data assimilation system accurately fits the measurements and estimated (most) emissions correctly. The emission differences maps per species between CTL – NAT_E and SUP_E – NAT_E are depicted FigureS 5.

We focus on the Sahara region and the estimated DU emissions to highlight an important issue of any data assimilation system for emission estimation. Figure 18 depicts the dust emission fluxes over the western Sahara for the NAT_E, CTL and SUP_E. Although the dust emission fields are similar, the spatial distribution of the dust sources differs. There are some grid cells where dust emissions are zero (not considered as sources by the model) in the control and the data assimilation experiment (highlighted with the red polygon at Figure 18d), while the same locations are active sources in the nature run. These differences are caused by the set-up of each dust scheme, where the preferential dust sources can differ (Schepanski et al., 2007). These contrasting assumptions can impact negatively the estimated emissions, since our data assimilation setup adjusts existing sources and does not introduce new sources. Dust emission differences for CTL – NAT_E (Figure 18d) shows an underestimation over these grid cells and the surrounding area in question. SUP_E – NAT_E  (Figure 18e) reveal that dust emissions remained underestimated over the same grid cells but the surrounding emissions (especially westward) were increased (overestimated) to compensate for the lack of dust in the area. Hence the data assimilation system did not only underestimate these specific grid cells but ended up overestimating all the surrounding area as well in order to compensate for the missing aerosol in the atmosphere. On the other hand, emission in areas where the location of preferential dust emission sources is the same, data assimilation did not have a problem estimating the correct emissions (highlighted with the orange polygon at Figure 18c). These examples show that it is possible for a data assimilation system to reduce source strengths in the model, while not possible (under the current dust scheme and data assimilation set-up) to start emitting dust in grid cells specified as non-sources. Consequently, dust schemes with spatially broader and continues sources may provide a more flexible

way to adjust the emissions based on observations. Note that although these examples reside in the modeling world of an OSSE, the same problem can be affecting the dust emission estimation of non-OSSEs data assimilation studies, since source location in models can differ from the source location in nature.

## 5 Conclusions

In this study we have quantified SPEXone ability to estimate aerosol emissions using a fixed-lag ensemble Kalman smoother (LETKS) in combination with the ECHAM-HAM aerosol-climate model. SPEXone is a passive remote sensing multi-angle polarimeter part of the NASA PACE missions scheduled to be launched at 2023. The system is tested using Observing System Simulation Experiments where the Nature Run is created by an ECHAM-HAM simulation with altered aerosol emissions from the standard model setup. Synthetic observations of aerosol optical depth, Angstrom exponent and single scattering albedo are

sampled from this nature run according to the spatiotemporal coverage of SPEXone or a theoretical sensor with almost perfect global coverage.

The data assimilation experiments based on SPEXone or the theoretical sensor provide similar results in terms of the estimated emissions and the simulated observations, which is very encouraging since it shows that SPEXone spatially limited observational coverage will be able to constrain emission almost as good as the theoretical satellite setup. Note that we assume

that 1.875 degree aggregate of SPEXone contain non-significant representation error, the observations of both sensors are unbiased and the differences in observations of the nature run and the data assimilation run are only caused by differences in emissions. We address most of these assumption by conducting additional experiments.

Specifically, the initial global prior emissions errors in the control run that ranged from 33% to 117% (depending on the species) drop to a range of 0% to 5% for the theoretical sensor and 0% to 11% for SPEXone. The highest difference between

the two sensors is observed on the SS estimated emissions mainly due to the lack of observations for SPEXone over India caused by cloudy conditions. An observational uncertainty scenario for SPEXone which doubles the uncertainty of the assimilated observations leads to reasonably good emission estimates. Further, we show the information of observations in days 5 and 6 after emissions is not that important for the estimation of emissions (for all species), but the information of observations in days 3 and 4 after dust emissions is very important and they should be used for the estimation of dust emissions.

Note that in all of these experiments the nature run was created using the same model and the same physics options as the data assimilation run, their only difference is that the emissions of the nature run were multiplied with emission factors globally constant and distinct for each aerosol species. Hence, the results of these data assimilation experiments may be too optimistic, since they do not account for any other uncertainty factor that would affect emissions estimation (e.g. meteorology biases, complexity in emission sources) in reality.

Therefore, additional experiments were conducted using the theoretical sensor in order to quantify the impact of other uncertainty factors that can affect the estimation of aerosol emissions. The role of biased meteorology is tested by nudging the wind components of the nature run to ERA-interim and the data assimilation run to ERA-5. Biased meteorology increases

mostly sea salt emissions global error in comparison to the data assimilation experiment where meteorology was not biased. The estimated emissions of the other species are negatively affected to a smaller extent.

Further, to investigate whether the creation of a Nature Run with emission scaling represents a too simple difference between nature and data assimilation run, an experiment where emissions in a new nature run are altered by changing the emission inventories and emission schemes. Data assimilation successfully reduced the global emission errors of all species, except for dust at some locations. Dust emission errors are not reduced because the preferential dust sources of the nature run are more compared to the data assimilation run. This complicates the emission estimation since dust is emitted from different location

in the nature run and the data assimilation run. Specifically in the western Sahara data assimilation increases dust emission extensively in its available dust sources based on the assimilated observations (sampled from the nature run) in order to compensate for the lack of dust that originated from dust sources only available in the nature run. This OSSE demonstrates that a data assimilation system may not provide the desirable results in case the location of emission sources are more sparse than nature.

This work highlights that the upcoming SPEXone sensor will provide high accuracy observations with sufficient coverage that contains information about the mass, size and absorption of the aerosol particles in order to estimate aerosol emission accurately using our data assimilation system. Using the full observational information of the PACE mission (SPEXone, HARP-2 and OCI), as well as using more retrieved aerosol properties (effective radius, refractive index) can potentially provide even better results.

**Appendix A: The effect of Prior Correction**

The Kalman filter assumes that the emissions do not have persistent errors, or in other words that the emissions are not biased (low or high) constantly in time. Unfortunately emissions in models can be biased, hence we developed a prior correction method to account for that. The effect of prior correction is tested by comparing the performance of the experiments with (SUP) and without (SUP0) prior correction. The simulated aerosols in the SUP0 experiment become almost identical to NAT,

although a small bias remained in all variables (FigureA 1). This is due to the setup of our OSSE, where the prior emissions of all the species are biased either low or high in comparison to NAT. In other words, although the uncertainty of prior emissions describe well the prior emission errors, the biased prior ensemble mean has a small toll on data assimilation performance. With prior correction (SUP) this issue is resolved and we get a better fit to the observations for all variables as shown in FigureA 1. The global error of the estimated emission is improved due to prior correction by 18% for SS and by up

to 7% for the other species (not shown). Although the effect of prior correction is small for SUP and SUP0, in the case where the prior emissions error differs a lot from the uncertainty of prior emissions, the effect of prior correction would be quite more significant, since it will adjust the ensemble mean of the emission perturbations and correct the bias of the model. An example of that is presented in subsection 4.3.2 for the estimate of SS emissions.

## Appendix B: SPEXone coverage based on a realistic ECHAM-HAM cloud mask

We want a realistic cloud mask that is nevertheless determined from the ECHAM cloud mask. The way we achieve this is by setting an ECHAM cloud fraction threshold, for all the grid cells that coincide with the cloud-free SPEXone spatiotemporal coverage. When ECHAM cloud fraction of a grid cell is lower that the cloud fraction threshold, we assume that at least some observations could be retrieved over the cloud free part of that grid cell. In order to make our results more realistic, we further change the cloud fraction threshold in each grid cell (in a statistical sense, by random draws) to make it appear more like

MODIS cloud mask.

Specifically, the grid cells of the cloud-free SPEXone mask were filtered out based on ECHAM cloud fraction greater than 0.7 (ECHAM-CloudMask1 red points in FigureA 2). Although ECHAM and MODIS cloud-based SPEXone masks almost matched in the total number of observations, they differed in the latitudinal and temporal distribution of observations (especially in high latitude and subtropics) (black and red points in FigureA 2). Thus, we allowed the 0.7 cloud fraction

threshold to change depending on how much the ECHAM and MODIS cloud-based SPEXone masks differ per latitude and time. This resulted to a SPEXone mask based on ECHAM cloud fraction but with the more realistic sampling that MODIS provides, specifically in time (ECHAM-CloudMask2 blue points in FigureA 2). The total number of observations retrieved by SPEXone based on MODIS and ECHAM cloud masks is depicted in FigureA 3.

## Appendix C: Observation uncertainty

We need to estimate the observational uncertainty for SPEXone, which is a sensor that is not yet launched. The retrievals errors of SPEXone are simulated as in Hasekamp et al. (2019a). The uncertainty of the retrieved parameters are propagations of uncertainties in both measured radiance/DoLP and the prior of the retrieved parameters .

Based on synthetic retrievals performed globally for four individual days, the standard deviation of the differences between the truth and the retrieved values were calculated for several $AOD_{550}$ classes. The results for $AOD_{550}$, $AE_{550-865}$ and $SSA_{550}$ are

shown in the FigureA 4. Note that relative differences where used for $AOD_{550}$ and absolute differences for $AE_{550-865}$ and $SSA_{550}$. Also for high $AOD_{550}$ cases where few retrievals were available, the uncertainty was calculated for $AOD_{550} > 1.6$ over land and $AOD_{550} > 0.8$ over ocean, to ensure that more than 50 cases were used in each case.

Over land retrievals have higher uncertainty than over ocean retrievals for almost all $AOD_{550}$ bands, in all variables. In addition retrievals in $AOD_{550} > 1$ have lower uncertainty than $AOD_{550} < 1$. The standard deviation of these relative and absolute

differences for each $AOD_{550}$ band were used to define the uncertainty of the assimilated observations for both the SUPER and SPEXone satellites. For example the uncertainty for the $AOD_{550}$ band 0.80 to 1.00 over land is 16.6% for $AOD_{550}$, 0.362 for $AE_{550-865}$ and 0.021 for $SSA_{550}$.

**Code and data availability**

The model simulations and the SPEXone simulated retrievals are available at the following link in zenodo: https://zenodo.org/record/5902137#.YfE4dPXMJ-U. The data assimilation software for aerosol emission estimation in ECHAM-HAM can be found in zenodo: https://doi.org/10.5281/zenodo.5596328. The ECHAM-HAM version that was used in this study can be found in the repository: https://svn.iac.ethz.ch/external/echam-hammoz/echam6-hammoz/branches/uni_amsterdam_vrije/ which can be accessed after registration at https://redmine.hammoz.ethz.ch/projects/hammoz. ERA-interim and ERA-5 are freely available in https://cds.climate.copernicus.eu/ after registration.

**Supplement**

The supplement related to this article is available online at: …

**Author contributions**

AT designed the experiments, with the help of NAJS and OPH, and carried them out. GF prepared SPEXone simulated retrievals. AT performed the analysis and prepared the manuscript with contributions from all co-authors.

**Competing interests**

The authors declare that they have no conflict of interest.

**Acknowledgements**

This work was carried out on the Dutch national e-infrastructure with the support of SURF Cooperative.

**Financial support**

This research has been supported by the Dutch Research Council (NWO) and Netherlands Space Office (NSO) (grant no. 2017.008). Athanasios Tsikerdekis is funded by a NWO/NSO project "AEROSOURCE: Estimation of Aerosol Emissions from Polarization Data" (grant no. ALWGO.2017.008).

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

**Table 1. Emissions** inventories **and schemes used per sector for all NAT experiments. Note that NAT and NAT_M use the same emissions inventories and schemes as CTL and DAS, but use emission factors (per species) to scale the emissions. ACCMIP: Atmospheric Chemistry and Climate Model Intercomparison. GFAS: Global Fire Assimilation System. CEDS: Community Emissions Data System.**

| Sectors | Species | CTL & DAS | NAT & NAT_M (emission factors) | NAT_E (schemes & inventories) |
|---------|---------|-----------|-------------------------------|-------------------------------|
| Dust | DU | ndust=5 | 0.5 | ndust=2 |

| | | | | |
|---|---|---|---|---|
| Sea Salt | SS | nseasalt=7 | 0.5 | nseasalt=2 |
| Oceanic | DMS | nseasalt=7 | 1.5 | nseasalt=2 |
| Forest Fires | OC, BC, $SO_2$, DMS | GFAS | 2 or 1.5 | ACCMIP |
| Grass Fires | OC, BC, $SO_2$, DMS | GFAS | 2 or 1.5 | ACCMIP |
| Domestic | BC, OC, $SO_2$ | ACCMIP | 2 or 1.5 | CEDS |
| Energy | BC, OC, $SO_2$ | ACCMIP | 2 or 1.5 | CEDS |
| Industry | BC, OC, $SO_2$ | ACCMIP | 2 or 1.5 | CEDS |
| Ships | BC, OC, $SO_2$ | ACCMIP | 2 or 1.5 | CEDS |
| Transport | BC, OC, $SO_2$ | ACCMIP | 2 or 1.5 | CEDS |
| Waste | BC, OC, $SO_2$ | ACCMIP | 2 or 1.5 | CEDS |
| Aircraft | BC | ACCMIP | 2 | CEDS |
| Agricultural waste burning | BC, OC, $SO_2$ | ACCMIP | 2 or 1.5 | ACCMIP |
| Biogenic | OC | AEROCOM II | 2 | AEROCOM II |
| Terrestrial | DMS | AEROCOM II | 1.5 | AEROCOM II |
| Volcanic (continuous) | $SO_2$ | AEROCOM II | 1.5 | AEROCOM II |
| Volcanic (explosive) | $SO_2$ | AEROCOM II | 1.5 | AEROCOM II |

**Table 2. List of experiments related to SPEXone.**

| Experiments | Satellite Coverage | Satellite Uncertainty | Add random error in observations | $\Delta T_a$, $\Delta T_s$ | Comments |
|---|---|---|---|---|---|
| CTL | × | × | × | × | No data assimilation |
| SUP | SUPER | SPEXone | × | 6, 2 | Data assimilation based on SUPER sensor (benchmark performance of the filter) |
| SPX | SPEXone | SPEXone | × | 6, 2 | Data assimilation based on SPEXone sensor |
| SPX_2U | SPEXone | SPEXone · 2 | × | 6, 2 | Data assimilation based on SPEXone sensor with double uncertainty |
| SPX_2URE | SPEXone | SPEXone · 2 | ✓ | 6, 2 | Data assimilation based on SPEXone sensor with double uncertainty and added random errors to the observations |
| SPX_W1 | SPEXone | SPEXone | × | 4, 2 | Data assimilation based on SPEXone sensor with shorter $\Delta T_a$ |
| SPX_W2 | SPEXone | SPEXone | × | 2, 2 | Data assimilation based on SPEXone sensor with even shorter $\Delta T_a$ |

**Table 3. List of experiments related to other uncertainty factors that can affect emission estimation.**

| Experiments | Assimilated Nature | Emission state vector | Prior Correction | Comments |
|---|---|---|---|---|
| SUP0 | NAT | by species | × | Test the effect of Prior Correction |
| SUP0_M | NAT_M | by species | × | Test the effect of biased Meteorology |
| SUP_E1 | NAT_E | by species | × | Test the effect of realistic emission differences between nature and data assimilation runs |

| SUP_E2 | NAT_E | by species | × | Test the effect of realistic emission differences between nature and data assimilation runs. Estimate emissions by mode for SS and DU |
|---|---|---|---|---|
| SUP_E | NAT_E | by species and mode | ✓ | Test the effect of realistic emission differences between nature and data assimilation runs. Estimate emissions by mode for SS and DU and enable prior correction without the prior max flag |

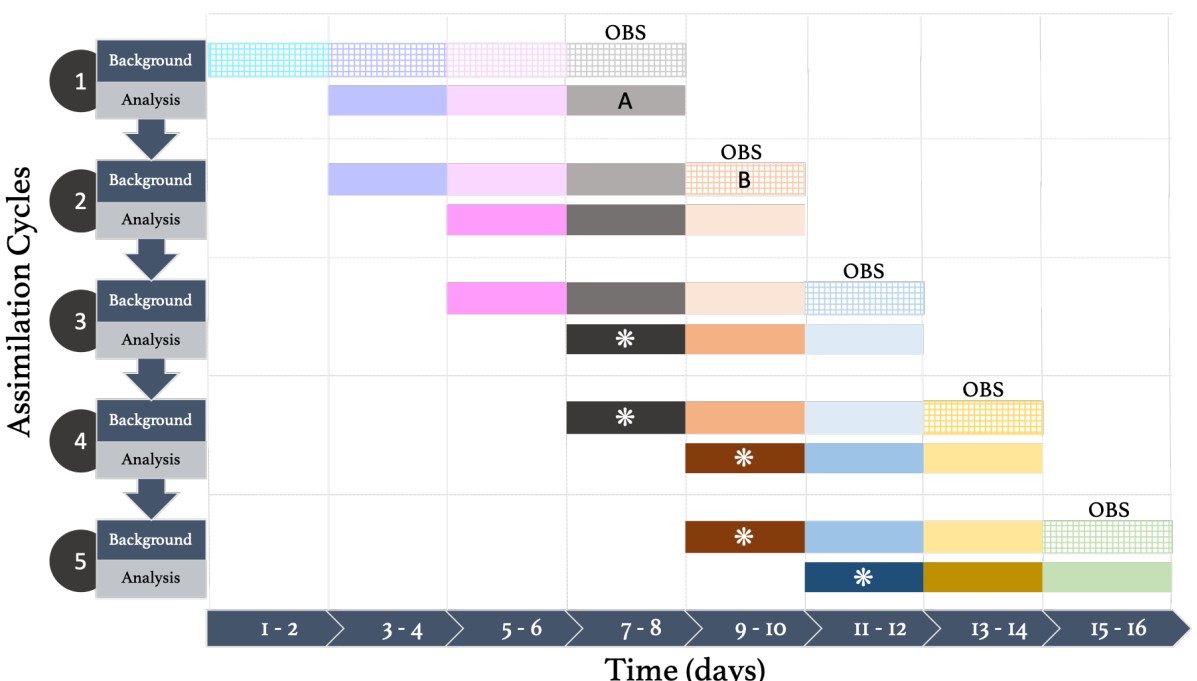

Figure 1. An illustration of the data assimilation system. The horizontal axis depicts time in segment of 2 days and the vertical axis the assimilation cycles, which of each is consisted by a background and an analysis step. Boxes are consisted by 32 spatially correlated perturbation maps for each perturbed parameter (DU, SS, OC, BC, SO$_4$, SO$_2$ and DMS) that are used to create the ensemble. Dashed coloured boxes indicate the default perturbations where the ensemble mean and standard deviation is equal to 1. Solid coloured boxes express the analysis emission perturbations that were affected by the assimilation of some observations. Solid coloured boxes with an asterisk (*) shows the posterior emission perturbations, corrected based on 6 days of observations. Different colours signify that every 2 days different perturbations are used. OBS indicate the assimilated observations for a 2-day period. A and B are marked in order to explain the prior correction (subsection 3.2).

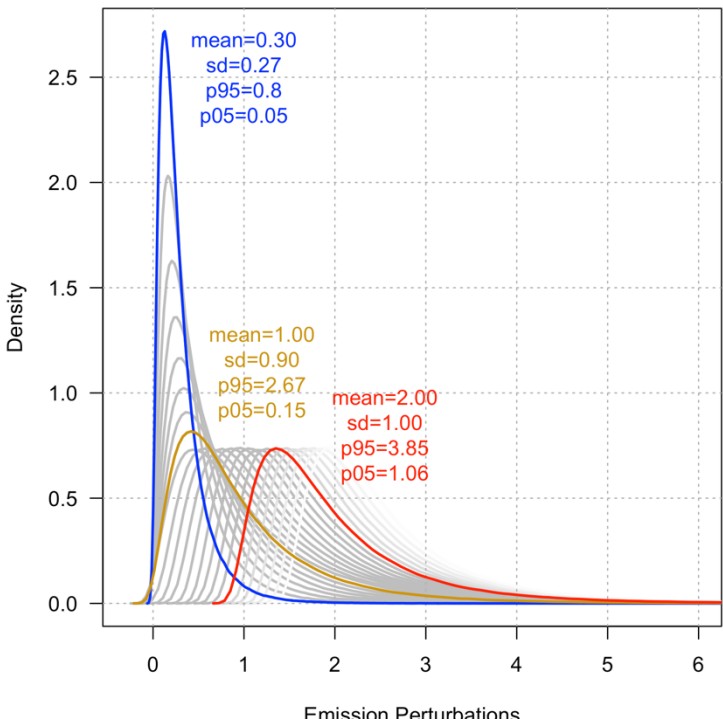

**Figure 2. An example that shows how the ensemble standard deviation (ENSSTD) is scaled according to the ensemble mean (ENSMEAN) with the prior correction option. Each distribution although it appears smooth for illustrative purposes, consists of 32 emission perturbation values, equal to our ensemble size. Blue, yellow and red curves highlight the statistics of three distributions with an ensemble mean of 0.3, 1 and 2 respectively. The 95% (p95) and 5% (p05) percentile indicate the approximate highest and lowest value of an ensemble member in these distributions. Grey curves represent in between distribution shapes (other than the ones highlighted) with different ensemble mean.**

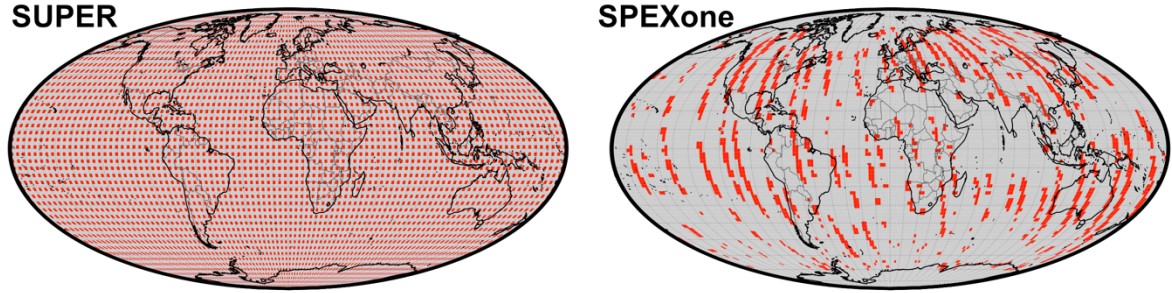

**Figure 3. Red grid cells illustrate the two day spatial coverage of SUPER and SPEXone instruments. SPEXone coverage is shown for the 17th and 18th August. In both cases each observation size is 1.875° x 1.875° (super observations) and includes estimates of $AOD_{550}$, $AE_{550-865}$ and $SSA_{550}$.**


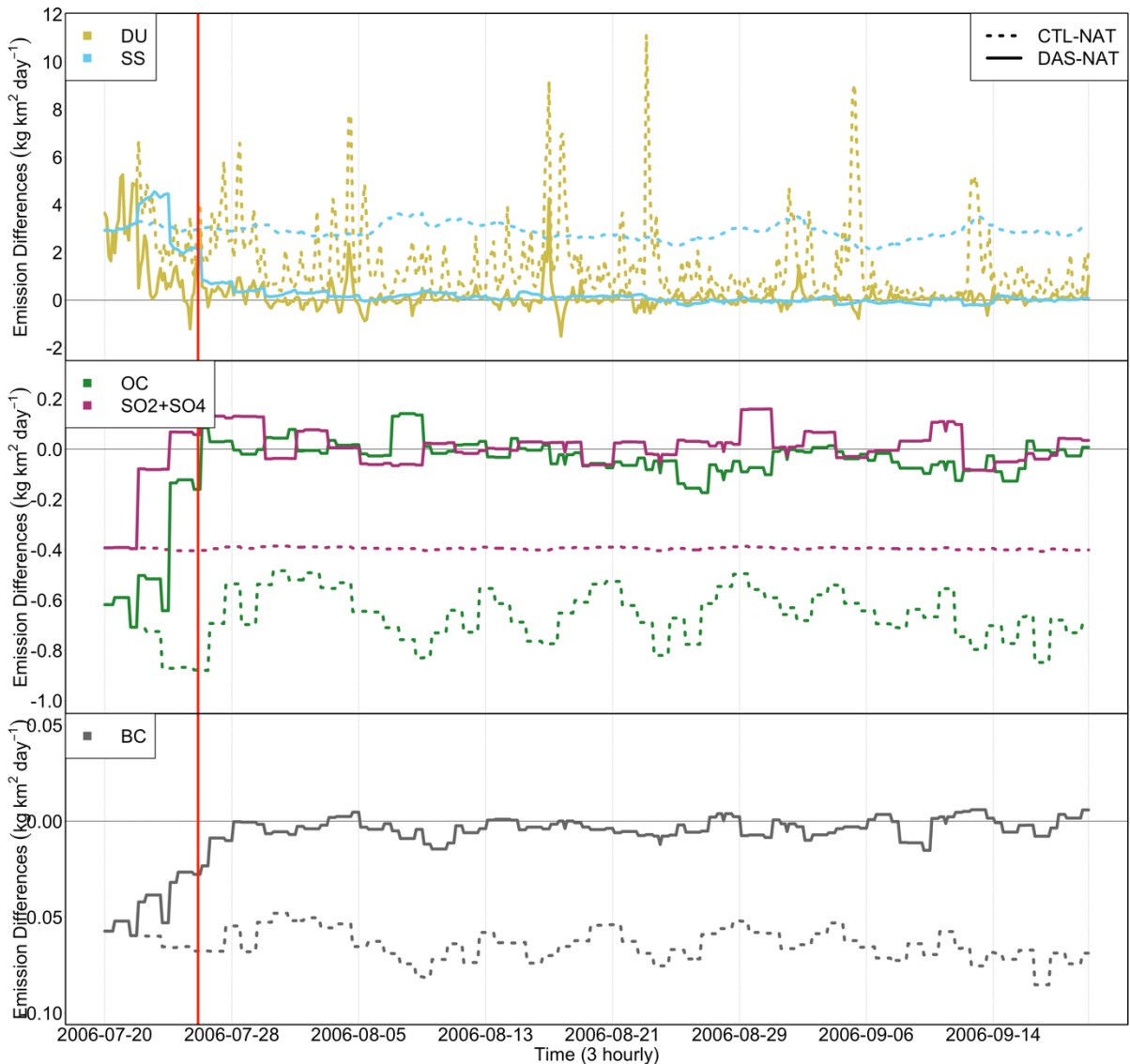

**Figure 4.** Timeseries for emission fluxes differences between CTL – NAT and SUP (DAS) – NAT for each species. Red line indicates the time where the analysis emissions perturbations were estimated for the first time. Note that $SO_4$ direct emissions are only a small fraction (2.5%) of $SO_2$ emissions in ECHAM-HAM, hence they are shown as a sum $SO_2+SO_4$ in the plot.

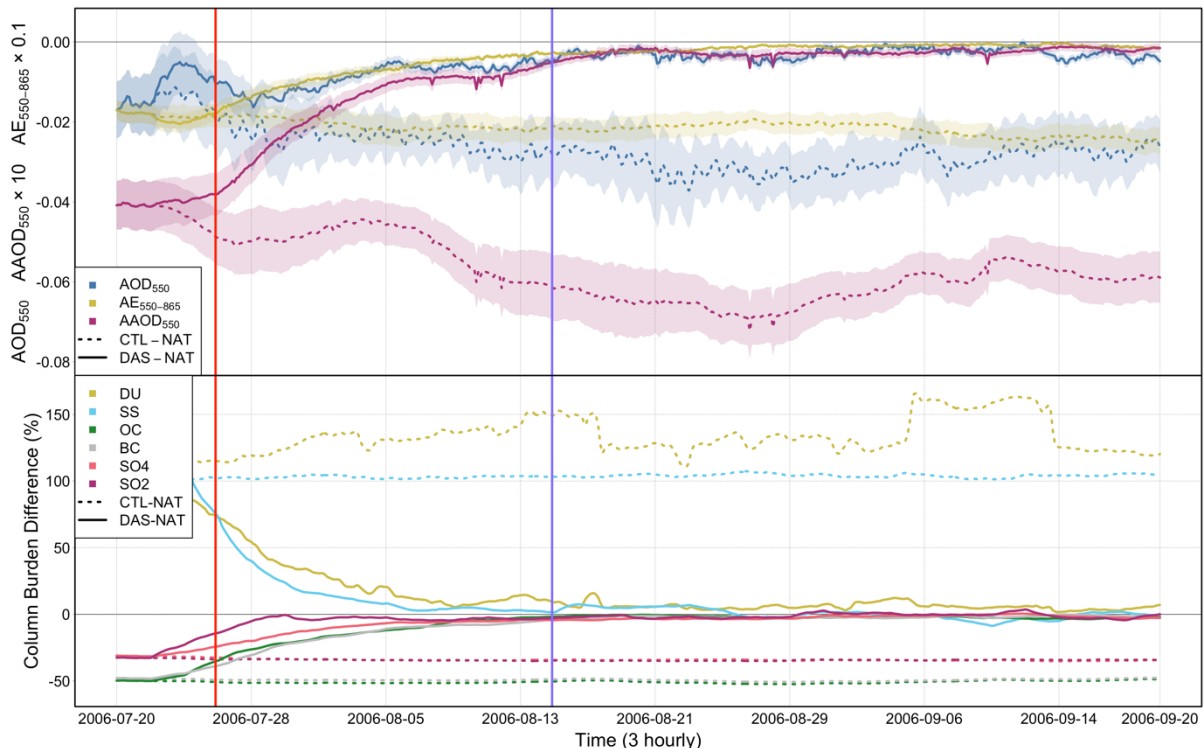

**Figure 5. Timeseries for aerosol optical properties differences and column burden relative differences between CTL – NAT (dashed lines) and SUP (DAS) – NAT (solid lines). Vertical red line indicates the time where the analysis emissions perturbations were estimated for the first time and vertical purple line indicates the time where the plotted variables reach equilibrium with the analysis emissions. The period between the red and the purple line indicates the lag time of global aerosol burden to react to the analysis emissions.**

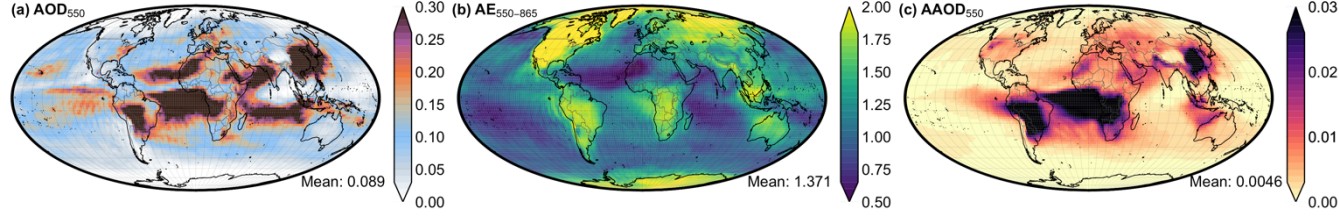


**Figure 6. Aerosol optical properties for the CTL experiment. Mean stand for the global mean value, estimated by averaging all the available grid cells.**

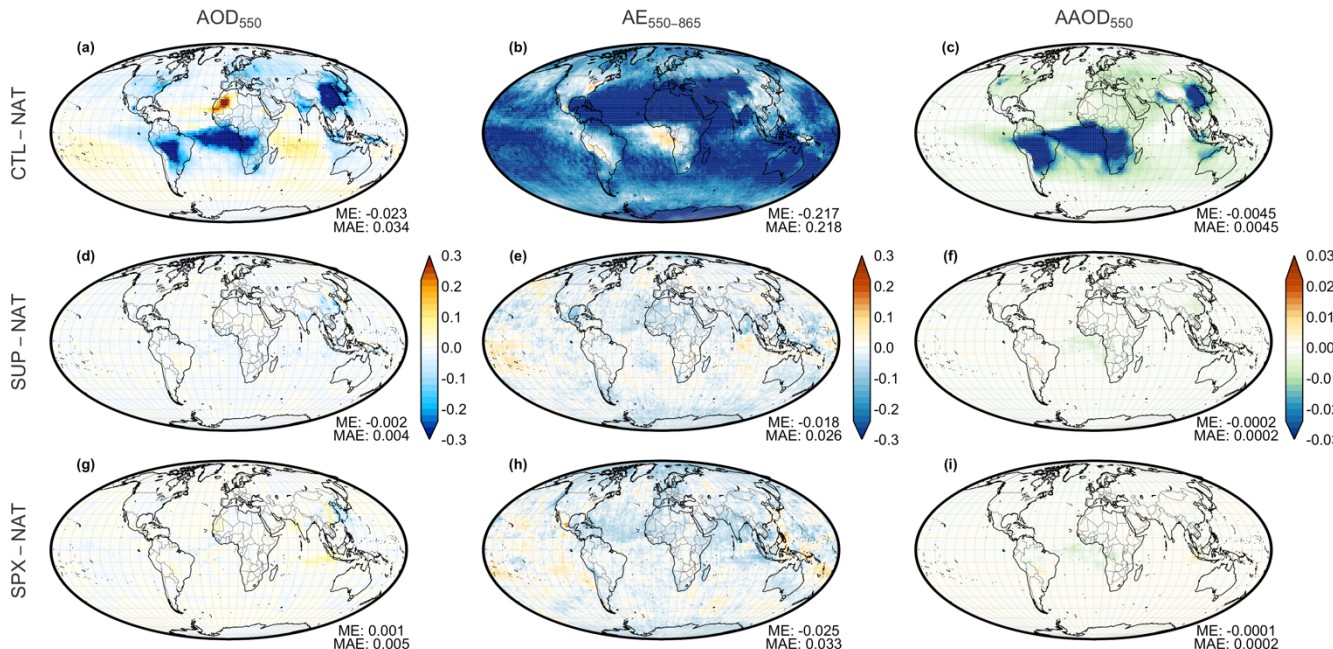

**Figure 7. Aerosol optical properties differences of CTL – NAT (a,b,c), SUP – NAT (d,e,f) and SPX – NAT (g,h,i). Left column depicts AOD (a,d), middle column AE (b,e) and right column AAOD (c,f).**

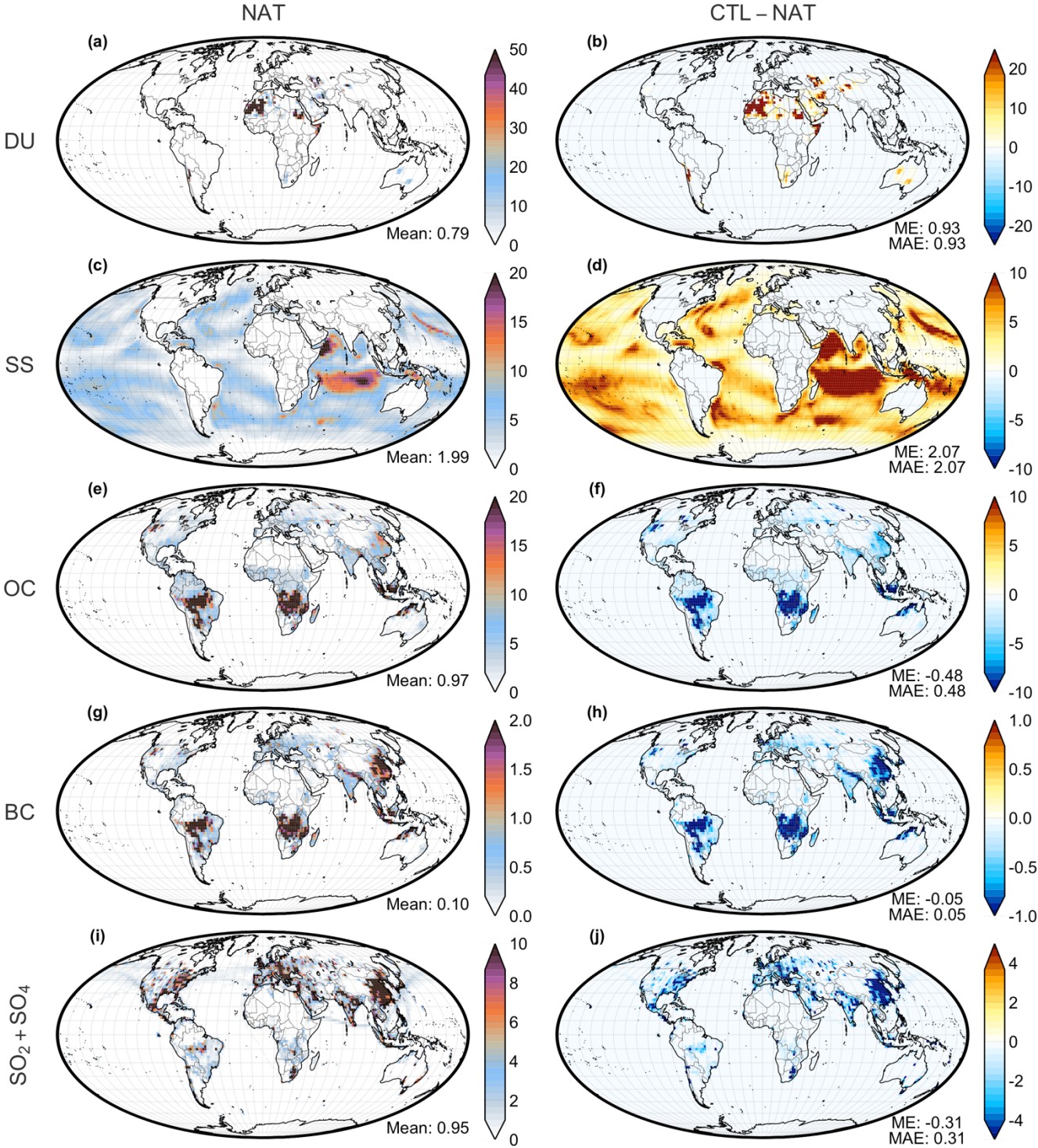

**Figure 8.** Aerosol emission fluxes (kg km$^{-2}$ day$^{-1}$) for NAT by species (a) DU, (b) SS, (c) OC, (d) BC, (e) SO$_2$+SO$_4$. The second column depict the differences between CTL − NAT.

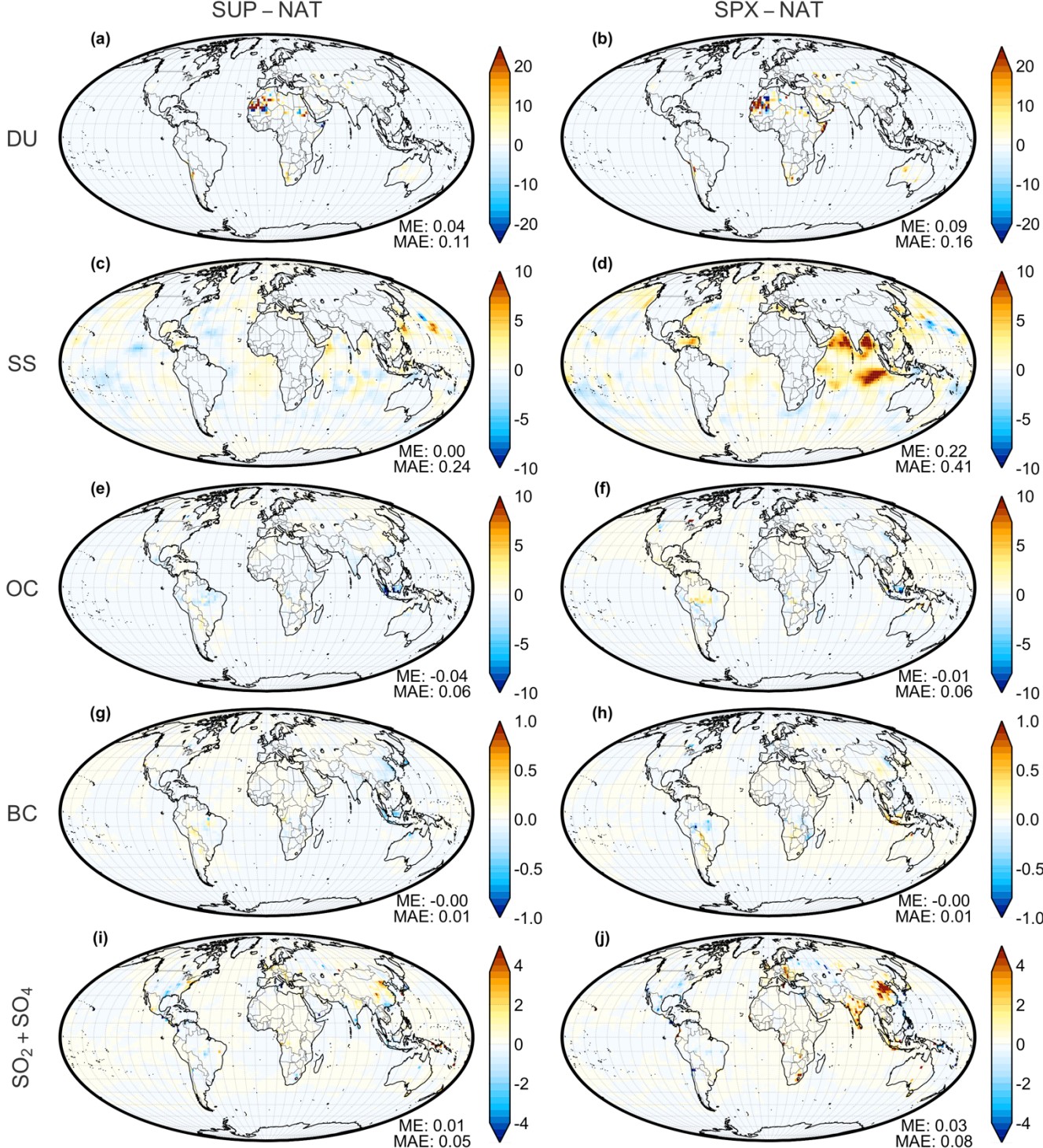

**Figure 9. Same as figure 8 but for the differences between SUP − NAT and SPX − NAT.**

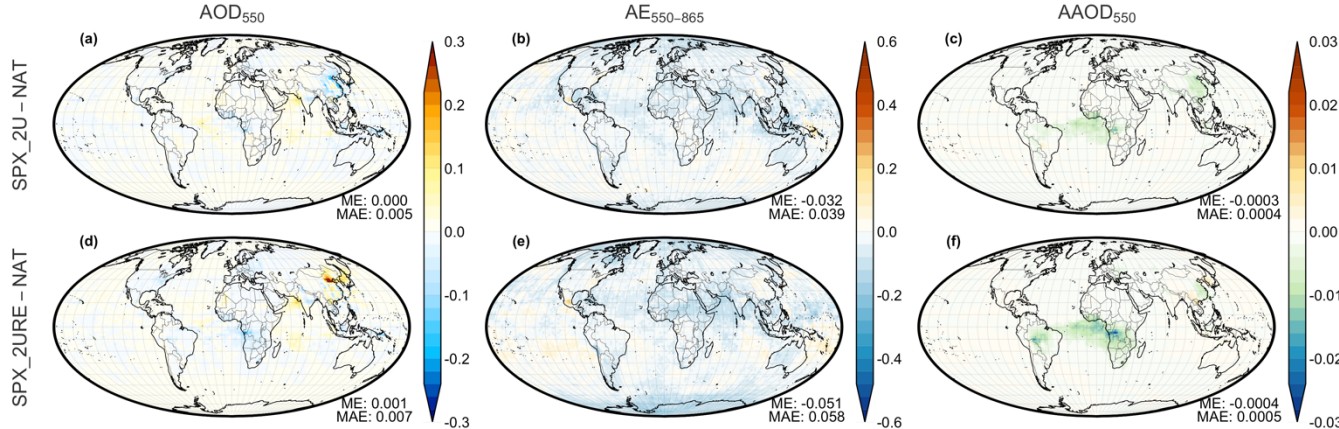

**Figure 10. Aerosol optical properties differences of SPX_2U – NAT (a,b,c), SPX_2URE – NAT (d,e,f). Left column depicts AOD (a,d), middle column AE (b,e) and right column AAOD (c,f).**

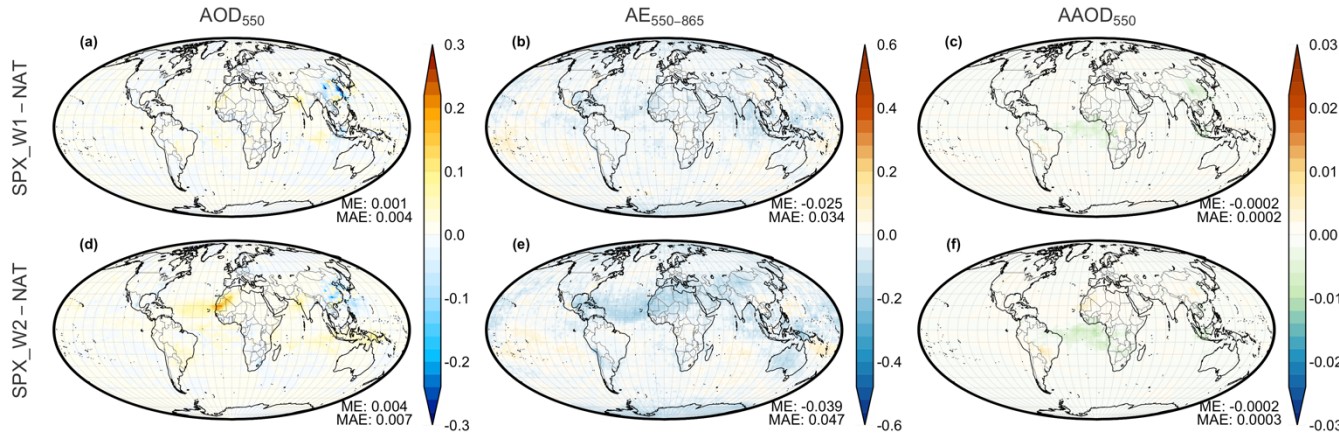


**Figure 11. Aerosol optical properties differences of SPX_W1 – NAT (a,b,c), SPX_W2 – NAT (d,e,f). Left column depicts AOD (a,d), middle column AE (b,e) and right column AAOD (c,f).**

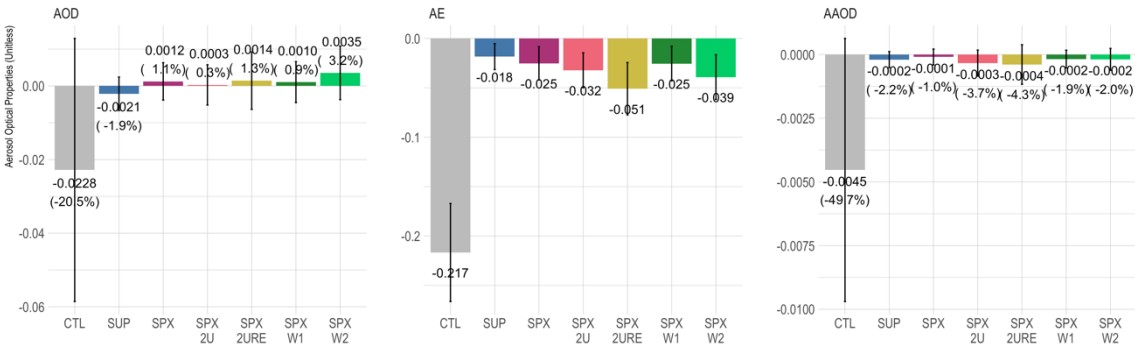

**Figure 12. Global mean differences of CTL and several data assimilation experiments from NAT. Parenthesis indicates the global mean relative difference. The error bar indicates the standard deviation of differences by grid for the whole globe. Large (small) error bar indicates that local differences are high (low).**

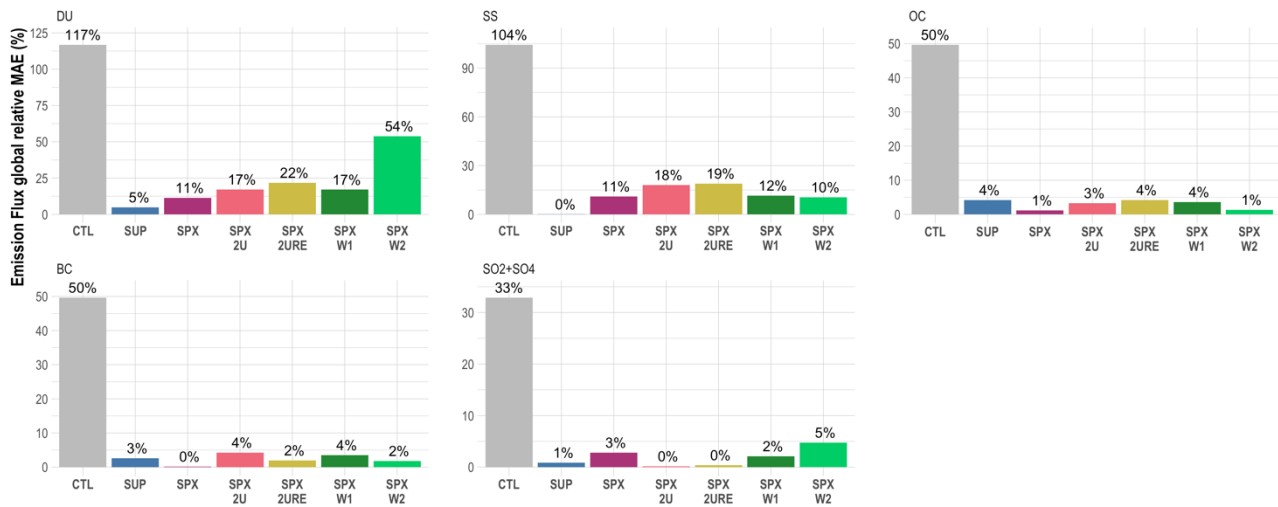

**Figure 13. Global relative MAE (%) of species-specific emission fluxes for several experiments.**

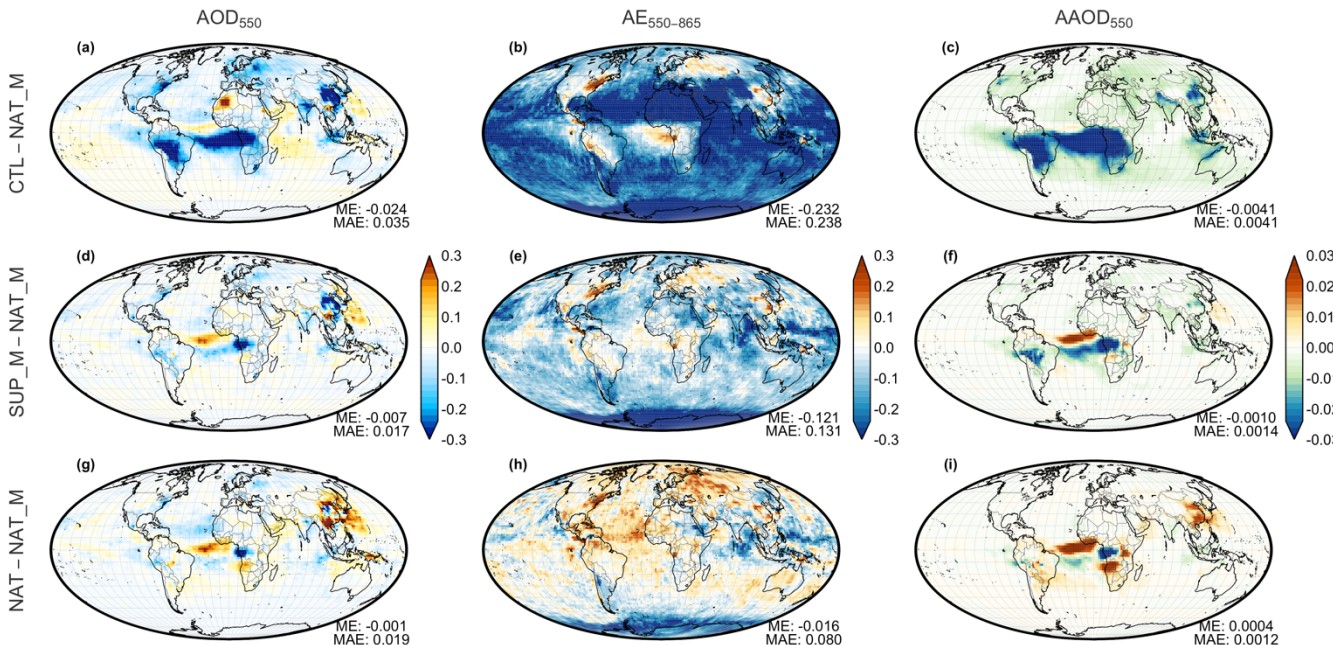

**Figure 14. Aerosol optical properties differences of CTL – NAT_M (a,b,c), SUP_M – NAT_M (d,e,f) and NAT – NAT_M (g,h,i). Left column depict AOD (a,d,g), middle column AE (b,e,h) and right column AAOD (c,f,i). Note that the differences of the last row indicate changes in aerosol optical properties only due to different meteorology.**

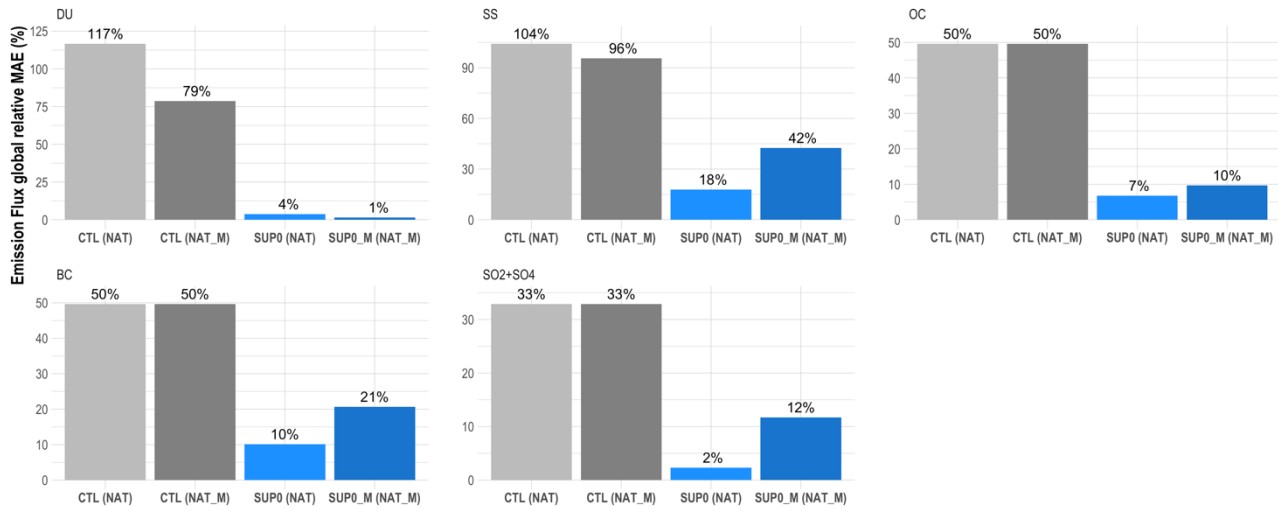

**Figure 15. Global relative MAE (%) of species-specific emission fluxes for several experiments. The parenthesis indicates the nature run which is used as a reference in each case.**

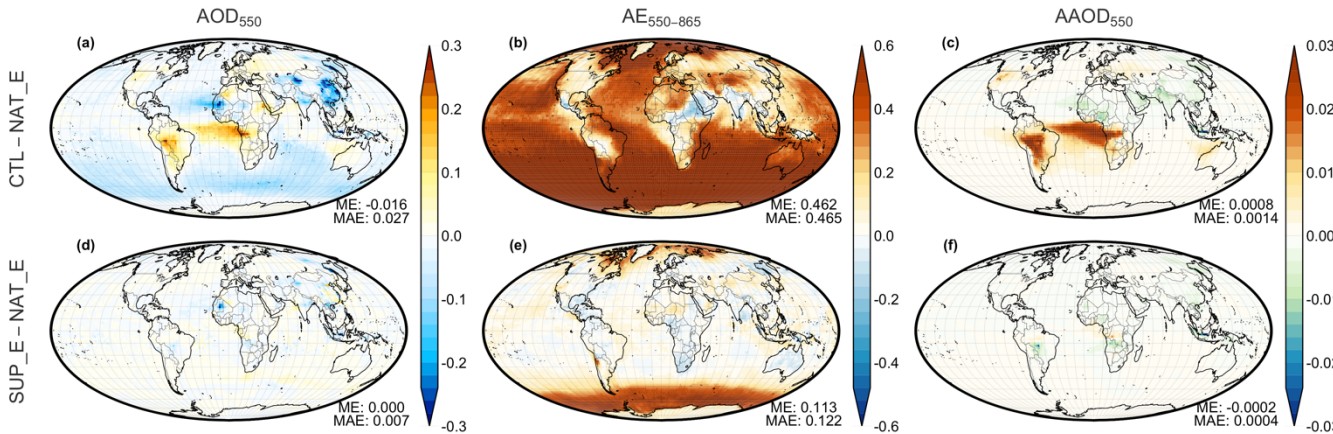

**Figure 16. Aerosol optical properties differences of CTL – NAT_E (a,b,c) and SUP_E – NAT_E (d,e,f). Left column depict AOD (a,d), middle column AE (b,e) and right column AAOD (c,f).**

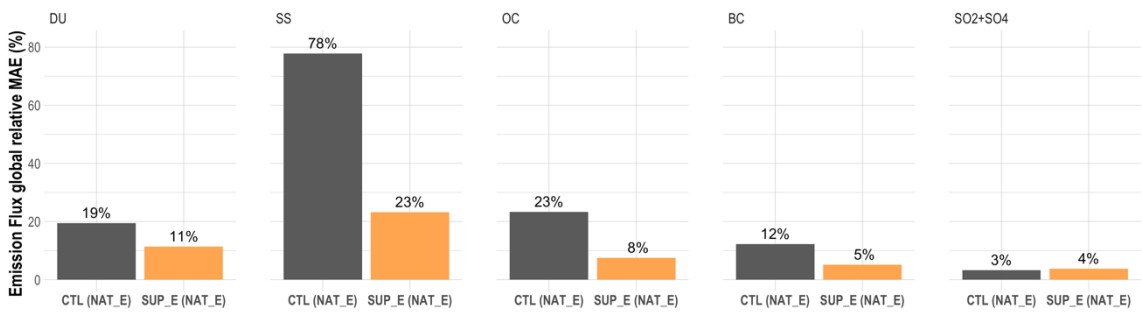

**Figure 17. Global relative MAE (%) of species-specific emission fluxes for several experiments. The parenthesis indicates the nature run which is used as a reference in each case. Note that statistics were calculated for sources that are active on NAT_E.**

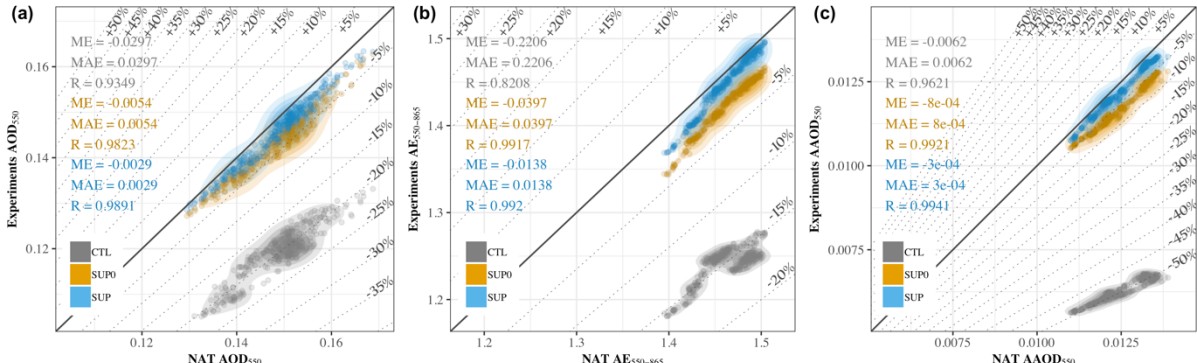

**Figure 18. Dust emission fluxes (kg km$^{-2}$ day$^{-1}$) for (a) the NAT_E, (b) the CTL and (c) the SUP_E. The differences between CTL – NAT_E and SUP_E – NAT_E are depicted at subplot (d) and (e) respectively. Note that NAT_E uses a different dust scheme than CTL and SUP_E, hence the location where dust can be emitted differs. In subplot (d), blue and red boxes highlight regions where dust emissions are overestimated and underestimated respectively in CTL compare to NAT_E. In the first case the data assimilation can modify the emissions and correct the overestimation, while in the second case it cannot (details in the subsection 4.3.2).**

**FigureA 1. AOD$_{550}$ (a), AE$_{550-865}$ (b) and AAOD$_{550}$ (c) scatterplot for NAT, SUP and SUP0 experiments. Each point represents a 3hourly global mean. ME: Mean Error, MAE: Mean Absolute Error and R: Pearson's Correlation. The shaded areas represents the 2D kernel density estimation for each experiment.**

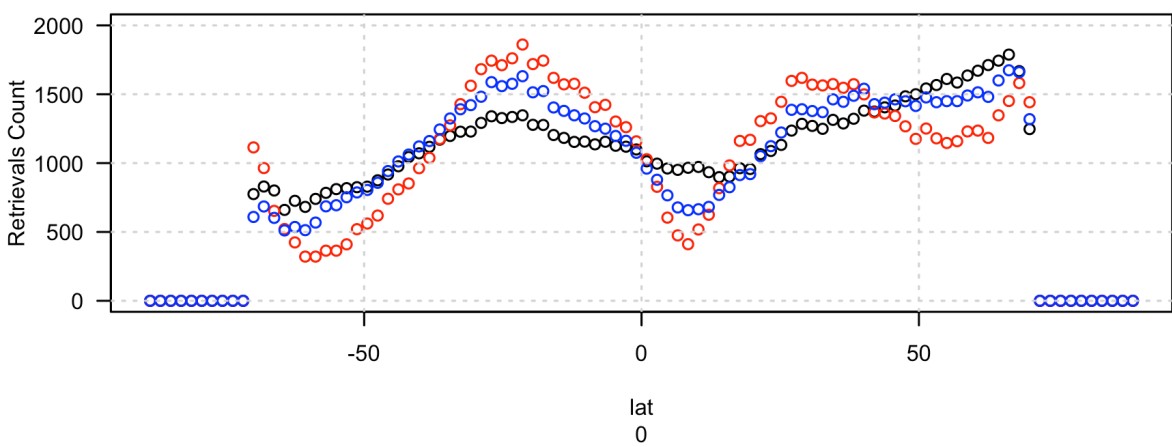

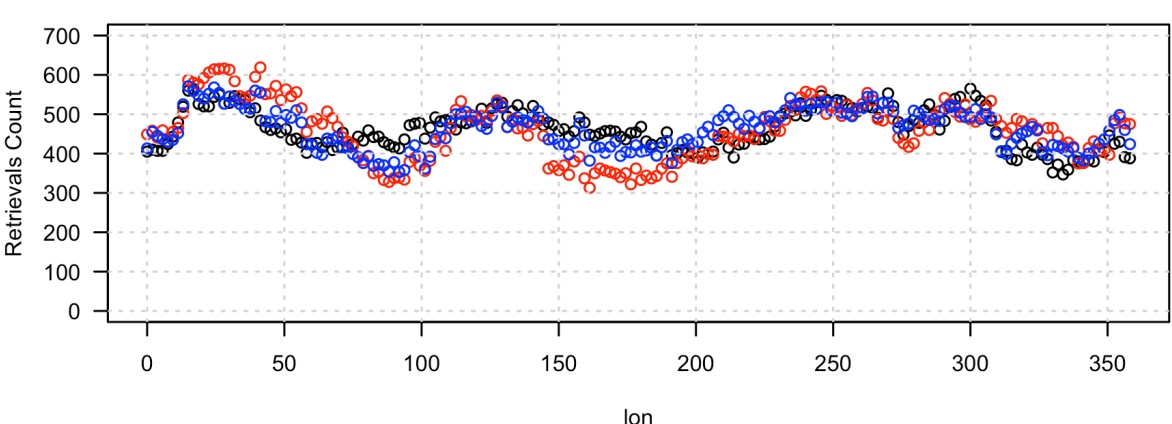

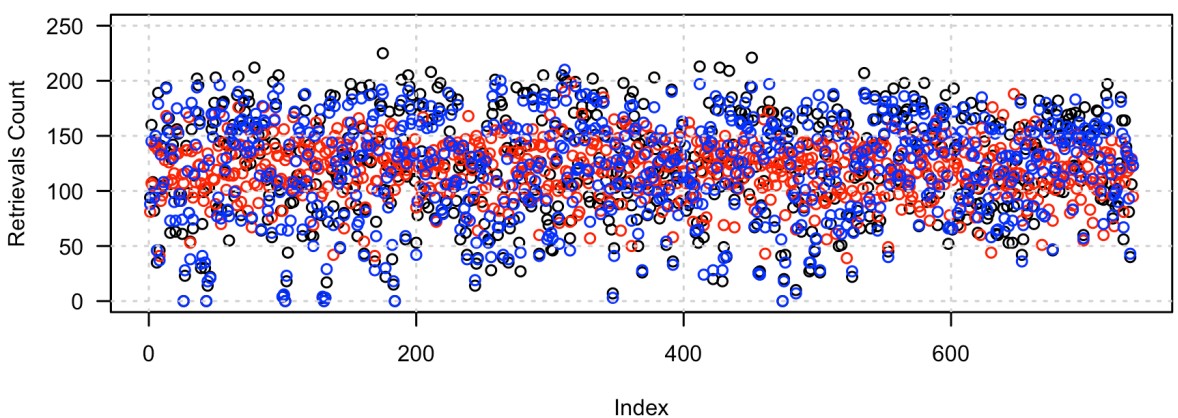

**FigureA 2. Number of observations by latitude, longitude and time for the SPEXone mask based on MODIS cloudiness (black; MODIS-CloudMask), ECHAM cloud fraction < 0.7 (red; ECHAM-CloudMask0) and ECHAM cloud fraction hybrid method explained in text (blue; ECHAM-CloudMask). The total number of observations for each mask is 88731 for MODIS-CloudMask, 88005 for ECHAM-CloudMask0 and 88886 for ECHAM-CloudMask. The analysis refers to the period July 2nd to October 1st.**

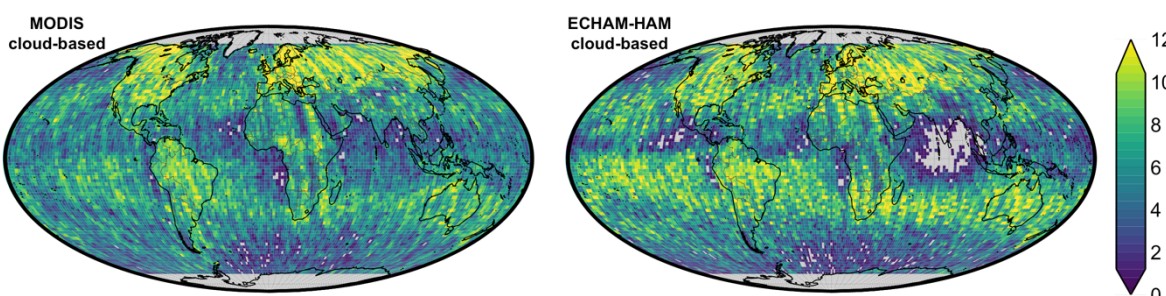

**FigureA 3. Number of observations for the MODIS and ECHAM-HAM cloud-based SPEXone masks. Each gridded observation includes an $AOD_{550}$, $AE_{550-865}$ and $SSA_{550}$ measurement. The analysis refers to the period July 2nd to October 1st.**

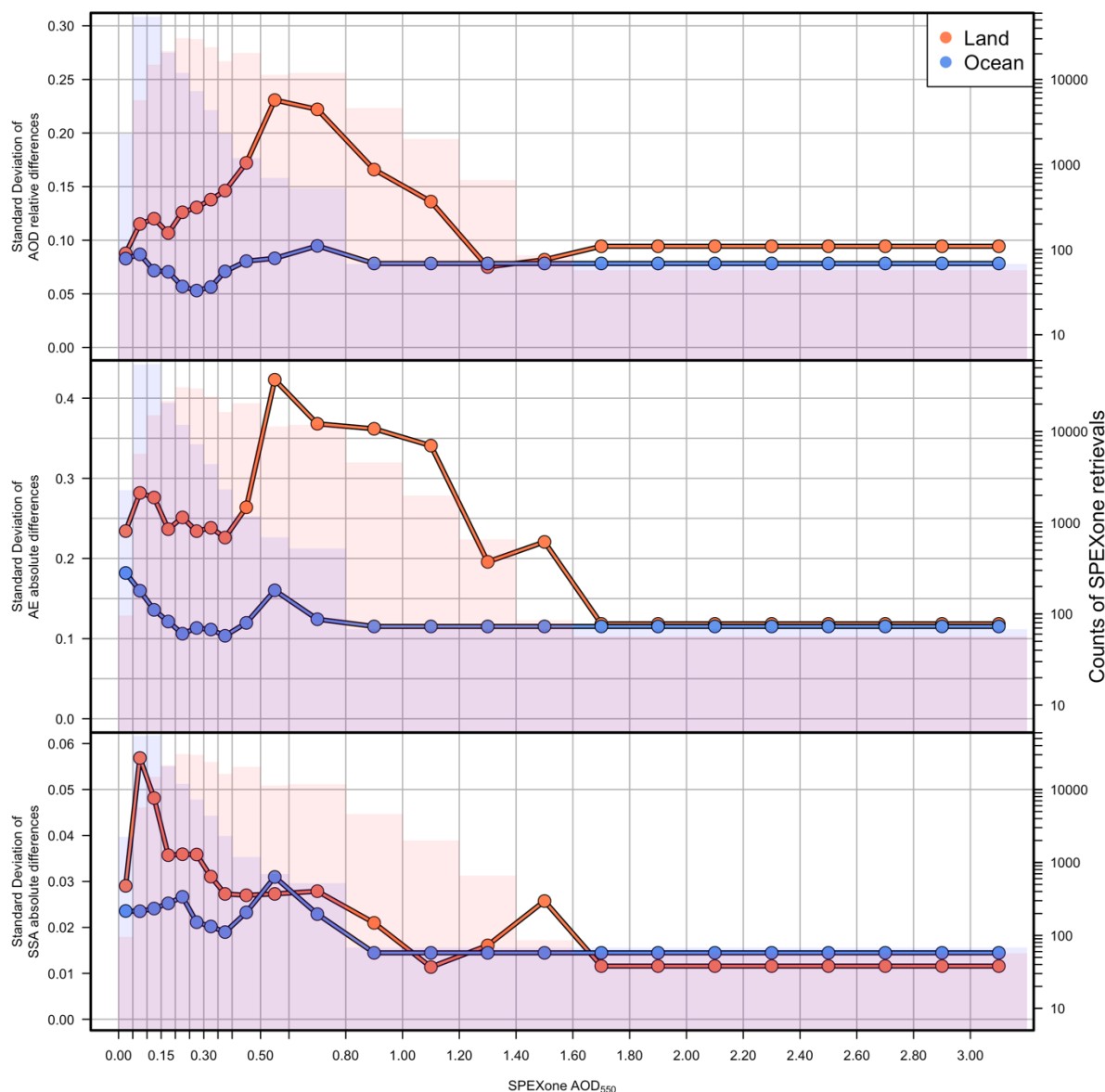

**FigureA 4. Defined uncertainty of SPEXone observations. Each point represents the standard deviation of the differences between TRUTH – RETRIEVED for a specified AOD$_{550}$ band. The analysis was carried out separately for retrievals over land and ocean. Bars depict the number of SPEXone retrievals for each AOD$_{550}$ classes and their height is associated with the right vertical axis.**