# Peer review of "Estimating aerosol emission from SPEXone on the NASA PACE mission using an ensemble Kalman Smoother: Observing System Simulation Experiments (OSSEs)"

_Geoscientific Model Development, 2021_

## Author Comment (AC1)

Response to Chief Editor for the manuscript: "Estimating aerosol emission from SPEXone on the NASA PACE mission using an ensemble Kalman Smoother: Observing System Simulation Experiments (OSSEs)"

Dear Chief Editor,

Thank you for highlighting these issues.

Regarding your first comment we have upload our model simulations and SPEXone data to the following link:
https://zenodo.org/record/5902137#.YfE3u_XMJ-U

Your second comment relates to the unsuitability of the SVN repository provided. According to the software license agreement (link below) of the Max Planck Institute for Meteorology (MPI-M) under subsection 4.2 I am not allowed to distribute the software. Specifically: "You are not allowed to distribute the "Software" in its original or modified form. Third party users must obtain the "Software" directly from MPI-M.".
https://mpimet.mpg.de/fileadmin/projekte/ICON-ESM/mpi-m_sla_201202.pdf

Indeed the access to the SVN repository we provide requires registration, which is given to any researcher that wants to work with ECHAM-HAM. Registering can be done by contacting the managers of the website:
https://redmine.hammoz.ethz.ch/projects/hammoz

Further the ECHAM-HAM branch in the SVN repository we provide will be kept frozen to the version used for this study.

The code availability was modified in the revised version of the manuscript to include all the new information:
**Code and data availability**
The model simulations and the SPEXone simulated retrievals are available at the following link in zenodo: https://zenodo.org/record/5902137#.YfE4dPXMJ-U. The data assimilation software for aerosol emission estimation in ECHAM-HAM can be found in zenodo: https://doi.org/10.5281/zenodo.5596328. The ECHAM-HAM version that was used in this study can be found in the repository: https://svn.iac.ethz.ch/external/echam-hammoz/echam6-hammoz/branches/uni_amsterdam_vrije/ which can be accessed after registration at https://redmine.hammoz.ethz.ch/projects/hammoz. ERA-interim and ERA-5 are freely available in https://cds.climate.copernicus.eu/ after registration.

Best regards,
Athanasios Tsikerdekis

---

## Author Comment (AC2)

Response to Referee #1 for the manuscript: "Estimating aerosol emission from SPEXone on the NASA PACE mission using an ensemble Kalman Smoother: Observing System Simulation Experiments (OSSEs)"

Dear Referee #1,

Thank you for reviewing our manuscript. Your comments definitely help to improve and define better some aspects of our work. Below you can find our point-by-point responses to all of your comments.

Best regards,
Athanasios Tsikerdekis

**Format**
Questions
Responses
"Quotes from the manuscript and revised or added text."

**Comments**
If I am not mistaken, the figures presenting the spatial distribution of the differences between the examined experiments present also as "mean" the mean difference (?) globally. Such an approach might result in masking of the error when positive and negative differences appear with similar frequency. I suggest also presenting the global mean of the absolute differences in order to have a more realistic overview of the differences between the experiments. Moreover, I suggest defining (in the text) the metrics used in the study (maybe as Appendix?).

Indeed the global mean error (ME) may end up very close to zero with regional positive and negative error that cancel themselves out (e.g. Fig 7a). Therefore, we have added along with the ME, the mean absolute error (MAE) in all global maps that depict differences, specifically Figure 7, Figure 8, Figure 9, Figure 10, Figure 11, Figure 14, Figure 16, FigureS 1, FigureS 2, FigureS 3 and FigureS 5. Additionally we have cited our previous publication where in Appendix B it contains all the metric used in the present study also.

"ME and MAE equations can be found in Appendix B of our preceding publication (Tsikerdekis et al., 2021)."

The NAT experiment represents the synthetic observations used for the assimilation. Do the authors have an estimate on how the NAT spatial variance is compared with that of the real observations? How does the comparison between the two might affect the improvement of emissions estimation? This is something that needs to be discussed in the text.

Thank you for this comment. Although these nature runs were created in order to test the data assimilation system capabilities and do not aim to represent the exact differences with a specific observational dataset (as indicated in subsection 3.3), it is interesting to test if the differences between CTL minus nature runs captures some of the general patterns of the differences between CTL minus an observational dataset. As an observational point of reference for this analysis we use retrievals from POLDER, since it could provide the same variables with SPEXone. We compare the difference between (i) CTL – POLDER, (ii) CTL – NAT (collocated over POLDER) and CTL – NAT_E (collocated over POLDER). We don't show the differences CTL – NAT_M, since they are pretty similar with the differences CTL – NAT.

The AOD underestimation over the biomass burning sources of Africa and South America as well as the overestimation over isolated ocean regions observed in CTL – POLDER is well

represented in CTL – NAT. Further the AE global overestimation and AAOD overestimation over South America observed in CTL – POLDER is captured by CTL – NAT_E. Independent of the sign of the differences, ME and MAE for the POLDER differences and the two nature run differences is comparable for all variables. Thus, this makes the nature runs of this work a fairly good proxy to represent some of the patterns illustrated in CTL – POLDER differences. Further the differences of CTL – NAT and CTL – NAT_E are contrasting in each of the variables, making them an ideal combination to test the emission estimation system under diverse scenarios. Further we have added the underlined sentence in main manuscript on subsection 3.3.

"These emission factors are chosen arbitrary, aiming to test if the data assimilation is able to estimate them correctly (test the system), rather than to reduce biases between NAT and a specific set of observations of an existing satellite (e.g POLDER-3). Nevertheless the differences between CTL – POLDER and CTL – NAT exhibit similarities in the biomass burning region in the Tropics and the global ME and MAE of these differences are on the same scale (not shown)."

[Figure]

**Aerosol optical properties differences of CTL – POLDER (a,b,c), CTL – NAT collocated over POLDER (d,e,f) and CTL – NAT_E collocated over POLDER (g,h,i). Left column depict AOD (a,d,g), middle column AE (b,e,h) and right column AAOD (c,f,i). Plots correspond to the period 15-08-2006 to 20-09-2006. Global Mean Error (ME) and global Mean Absolute Error (MAE) is depicted in the right bottom corner of each map.**

**Minor Comments**

P2, L33 and where applicable: I suggest using the Oxford comma as "size, and absorption with..".

Thank you for your suggestion. The lack of Oxford comma is consistent throughout the manuscript. Since this is a matter of preference according to the guidelines of GMD (as long as is consistent), we kept it as it is.

P2, L42: Add a comma after "In addition".
Added.

P2, L50: the have largest -> have the largest
Done.

P4, L104: Add angle units

Added.

P4, L105: that it measures -> measuring
Changed.

P4, L106 and where applicable: Add space before nm in 700nm.
Added.

P4, L107. Multiple sentences here starting with SPEXone. Please rephrase
Thank you for noting it. We have rephrased the three sentences.

P5, L139: taken into account -> are also taken into account
Corrected.

P11, L330: I suggest replacing the first sentence with "Figure 4 shows that the differences between DAS and NAT (solid lines) reach a value close to zero after 26 days." In this case you can delete "(Figure 4)" in the nest sentence.
Changed as suggested.

P12, L357: "nature run" Please use the abbreviations (NAT here) throughout the manuscript
Corrected.

P12, L386: from NAT -> from NAT for AOD, AE, and AAOD.
Corrected.

Figure 2: Maybe replace purple with green to better distinguish from blue.
Changed purple with dark yellow. The green and red combination is bad for people with the most common color deficiency (deuteranopia).

Figure 6 caption: Please indicate what the presented Mean value stands for.
Added in caption: "Mean stand for the global mean value, estimated by averaging all the available grid cells."

Captions of Figures 10 and 11: depict -> depicts
Corrected.

Figure 13: The experiments name is missing from the DU and SS figures.
Added.

Figure 18c: I suggest replacing orange rectangle with a blue one to better distinguish from the red one. Please also describe what the rectangles stand for in the respective caption.
Corrected the colors in the plot and made the boxes thicker. Also added the following in caption: "In subplot (d), blue and red boxes highlight regions where dust emissions are overestimated and underestimated respectively in CTL compare to NAT E. In the first case the data assimilation can modify the emissions and correct the overestimation, while in the second case it cannot (details in the subsection 4.3.2)."

Figure A 1: There seem to be some color shade areas in the scatter plot. Please indicate in the caption what they stand for.
Thank you for noting this. We added on the caption: "The shaded areas represents the 2D kernel density estimation for each experiment."

---

## Author Comment (AC3)

Response to Referee #2 for the manuscript: "Estimating aerosol emission from SPEXone on the NASA PACE mission using an ensemble Kalman Smoother: Observing System Simulation Experiments (OSSEs)"

Dear Referee #2,

Thank you for reviewing our manuscript. Your comments definitely help to improve the readability and define some aspect of our work better. Below you can find our point-by-point responses to all of your comments.

Best regards,
Athanasios Tsikerdekis

**Format**
Questions
Responses
"Quotes from the manuscript and revised or added text."

**Main Comments**
This paper describes an ensemble of OSSEs to estimate the top-down constraints on aerosol emissions based on observations provided by SPEXone. While the experimental setup and the scientific results are interesting, the paper needs improvement before publication. In particular, the english should be checked throughout. Additionally, the text needs to be more concise and some sections are unclear.

Thank you for your comments and appreciate your criticism. We have reread and improved the manuscript where we deemed necessary. Specifically we have given special attention to the subsections you mentioned on you minor comments (Abstract, Subsection 3.1 and Subsection 3.2).

**Minor Comments**
Abstract: please shorten the abstract, which currently reads more like a conclusion.

Indeed the abstract was very detailed and we shorten it considerably from 483 words to 381 words (reduced by 20%). We hope the new version gives a more concise and laconic summary of our work.

L168-170: The use of the term analysis for 1 cycle and posterior for several ones here makes little sense. Please find another terminology.

In our work emissions are estimated iteratively in time. Which means that the estimated emissions are going to be affected from observations on the same day, but also by observations in subsequent days. Therefore the term analysis is reserved for updated emissions that were corrected by n days of observations (n < $\Delta T_a$), while posterior for updated emissions that were corrected based on the full length of the smoother lag (n = $\Delta T_a$). This can be better understood visually in Figure 1. We realize that the sentence may have caused confusion so we have rephrased it:

"Note here that the term analysis is used to indicate the updated emissions affected by n days of observations (where n < $\Delta T_a$), while the term posterior is used to indicate updated emissions affected by $\Delta T_a$ days of observations (Figure 1)."

L177: And what impact did you find?

The impact is discussed in detail on the result section and specifically on subsection 4.2 Emission estimation using SPEXone – Sensitivity experiments.

The impact on the optical properties: "SPX_W1 and SPX_W2 reduce the $\Delta$Ta length to 4 and 2 days (from 6), hence less observations are used to derive the analysis emissions in each assimilation cycle and only 2 and 1 assimilation cycles (instead of 3) are used to calculate the analysis emission perturbations. The results reveal that $\Delta$Ta=4 days (SPX_W1) is sufficient to constrain the AOD, AE and AAOD in a similar manner as a $\Delta$Ta=6 days (SPX) (Figure 11 a,b,c). In other words, under the current experimental setup, observations 5 to 6 days after the emissions probably hold very little information for the correction of these emissions, and their exclusion has a very limited impact on the data assimilation performance. Contrary the experiment SPX_W2 shows a degradation in performance over western Sahara and North Atlantic for AOD and AE (Figure 11 d,e,f), indicating that observation in subsequent days 3 and 4 hold useful information for the correct estimation of emissions at day 1 and 2 as discussed in the next paragraphs. Note that SPX_W1 and SPX_W2 need ~33% and ~66% less computational resources than SPX respectively, since the background step in each assimilation cycle is shorter."

The impact on emissions: "Finally, SPX_W1 emission bias increases no more than 6 percent points in comparison to SPX in all species. However, dust emission error grows to 54% in SPX_W2 from 17% in SPX_W1, indicating that the information content of observations 3 and 4 days after the emissions is very rich and it should be used to correct these emissions, especially for Sahara dust plumes that extent over the Atlantic Ocean and last for several days. The emissions of OC, BC and SO2+SO4 are estimated very accurately by all of the data assimilation experiments, with relative MAE ranging from 0% to 5%, which indicates that in terms of the global mean emission estimation these emissions are unaffected by the sensor spatial coverage and observational uncertainty increase that were tested."

L187: "[...] a unique distribution to drive the emissions[...]"<-> "has a distinct prior emission distribution"
Changed as suggested.

L188: Should remove that sentence. This is confusing. What is the approach used to generate the prior error correlations? Please describe clearly.
Thank you for giving us the chance to clarify. We have rephrased the sentence and provided a reference to our previous paper where we describe this process in detail.
"Changes in neighboring grid cells of each member are not abrupt but smooth. This spatial correlation of the prior perturbations was generated using spatial smoothing, a method where data points are averaged with their neighbours. A step by step description on how our spatially correlated perturbations are created can be found at subsection 3.2 of our preceding work (Tsikerdekis et al. 2021).

L193: The mean and the standard deviation of the distribution (or the ensemble), not of the perturbations
Replaced "perturbations" with "distribution", as suggested.

L195: Please better explain the rationale here.
Indeed a more detailed explanation was lacking at this point. Thus we have added:
"Furthermore, it is noted that the perturbations are uniquely defined every $\Delta$Ts=2days (different colors in the boxes of Figure 1). The rationale here is that the simulated observations and emissions at day D (where D is any integer number) will be more correlated than the simulated observations at day D+$\Delta$Ts and emissions at day D.

Consequently, changes in emissions caused by assimilated observations of day D will be stronger compare to changes in emissions by assimilated observations of day D+$\Delta$Ts. This design is based on the fact that observations on the day of the emissions carry more information about the emissions, than observations in subsequent days."

Section 3.2: The whole section is not clear. It needs to be rewritten entirely. Please utilize equations rather than long sentences wherever appropriate. This will greatly facilitate the reading and the understanding of the so-called "prior correction" approach (currently unclear).

Thank you for noting this. We have adjusted subsection 3.2 making it more concise and adding equations where it was necessary. Further, we highlight that Figure 1 and Figure 2 improves the readability and comprehension of this subsection as they were designed with special attention to the details referred in the text.